



# Mesoscale observations of temperature and salinity in the Arctic Transpolar Drift: a high-resolution dataset from the MOSAiC Distributed Network

Mario Hoppmann[1], Ivan Kuznetsov[1,*], Ying-Chih Fang[2,*], and Benjamin Rabe[1,*]

[1]Alfred-Wegener-Institut Helmholtz-Zentrum für Polar- und Meeresforschung, Bremerhaven, Germany
[2]National Sun Yat-sen University University, 80424 Kaohsiung, Taiwan
[*]These authors contributed equally to this work.

**Correspondence:** Mario Hoppmann (Mario.Hoppmann@awi.de)

**Abstract.**

Measurements targeting mesoscale and submesoscale processes in the ice-covered part of the Arctic Ocean are sparse in all seasons. As a result, there are significant knowledge gaps with respect to these processes, in particular related to the role of eddies and fronts in the coupled ocean–atmosphere–sea ice system. Here we present a unique observational dataset of
upper ocean temperature and salinity collected by a set of buoys installed on ice floes as part of the Multidisciplinary drifting Observatory for the Study of Arctic Climate (MOSAiC) Distributed Network. The multi-sensor systems, each of them equipped with five temperature and salinity recorders on a 100 m long inductive modem tether, drifted together with the main MOSAiC ice camp through the Arctic Transpolar Drift between October 2019 and August 2020. They transmitted hydrographic in situ data via the iridium satellite network at 10 minute intervals. While three buoys failed early due to ice dynamics, five of them
recorded data continuously for 10 months. Four units were successfully recovered in early August 2020, additionally yielding internally stored instrument data at 2 minute intervals. The raw datasets (Hoppmann et al., 2021i) were merged, processed, quality-controlled and validated using independent measurements. Upon acceptance of the manuscript, the finally processed dataset (currently under moratorium) will be made publicly available under Hoppmann et al. (2022i). As an important part of the MOSAiC physical oceanography program, this unique dataset has many synergies with the manifold co-located observational
datasets, and is expected to yield significant insights into ocean processes, and to contribute to the validation of high-resolution numerical simulations.

## 1 Introduction

Oceanic mesoscales host a variety of important features and processes that have been extensively studied during the past four decades, pioneered by the Mid Ocean Dynamics Experiment in the 1970s (MODE, Bretherton et al., 1976; Jochum and
Murtugudde, 2006; McWilliams, 1976). Mesoscale processes are characterized by horizontal scales of few to several tens of kilometres and temporal scales much greater than the inertial periods. An important consequence is that the vertical velocity is $10^3$ to $10^4$ times weaker than the horizontal velocity (Thomas et al., 2008). The mesoscales are the major contributors in terms



of the kinetic energy reservoir (Ferrari and Wunsch, 2008) and usually manifest in the form of geostrophic eddies that are shown ubiquitously in present-day satellite imagery (McGillicuddy, 2016). Their occurrence has been linked to concurrently occurring

tilted isopycnals (Gill et al., 1974), where baroclinic instabilities trigger the release of potential energy which generates eddies. Global estimates of mesoscale eddy variability indicates that mesoscale eddies can be correlated with the first mode baroclinic Rossby radius (Chelton et al., 1998; Smith, 2007). In the Arctic Ocean, recent studies have highlighted the role of mesoscale eddies and the significant fraction that eddy kinetic energy contributes to total kinetic energy in observations (Zhao et al., 2014) and high-resolution numerical simulations (Wang et al., 2020). At the same time, it is likely that eddies have been historically

underestimated in key regions of the Arctic Ocean (Porter et al., 2020).

Apart from the mesoscales, recent advancements of instrumentation and computing power have allowed us to appreciate further the submesoscale regime, which resides between the mesoscale and turbulence (Mcwilliams, 2016). The submesoscales, by contrast, have the O(1) Rossby and Richardson numbers, and evolve with relative shorter spatial O(100 m to 10 km) and temporal O(days) scales. Most importantly, their non-negligible vertical velocities (D'Asaro et al., 2011; Thomas et al., 2008)

make them unique in bridging the momentum exchange between the ocean surface layer and the interior underneath (e.g. Lévy et al., 2018). Numerical simulations have shown that shifts from mesoscale to submesoscale regimes (Capet et al., 2008) result in frontal instabilities (e.g. Mahadevan, 2016; Mahadevan et al., 2010) and secondary ageostrophic circulation (Thomas, 2005).

This circulation leads to significant vertical velocity (Biddle and Swart, 2020; Mahadevan et al., 2010) advecting interior properties such as nutrients and salt into the productive surface mixed layer. Fadeev et al. (2021) and Kaiser et al. (2021) have

shown that in the Arctic Ocean, these features can shape distinct biogeochemical conditions and microbial as well as zooplankton communities, underpinning their underestimated role for surface ocean biodiversity and biogeochemical cycles. Recent observational studies also indicated that submesoscale processes are responsible for springtime upper-ocean restratification (du Plessis et al., 2019). At the same time, this is also the scale that is beyond the resolution of typical ocean general circulation models. It has been shown that the inclusion of submesoscale dynamics will fundamentally impact outputs of numerical sim-

ulations, and their covariance with inherent mesoscale dynamics still remains an open question (e.g. Lévy et al., 2012). Thus, appropriate parameterizations regarding submesoscale processes are still under development.

In the Arctic Ocean, submesoscale features have been observed that lead to the conversion of potential to eddy kinetic energy by slumping fronts (Zhang et al., 2019) or submesoscale filaments in the marginal ice zone (MIZ) with feedback between ice and ocean currents (von Appen et al., 2018). Further efforts used numerical models focused on frontal instabilities (e.g.

Manucharyan and Timmermans, 2013) and the MIZ (e.g. Manucharyan and Thompson, 2017). In the Arctic basin, mesoscale and submesocale variability does not only occur under seasonal ice cover and in the MIZ (e.g. Gallaher et al., 2016), but also under perennial sea ice. Although eddy activity is lower in the central Arctic than near the boundaries, the portion of total kinetic energy due to mesoscale eddies (eddy kinetic energy) is significant in the central Arctic Ocean (Zhao et al., 2014; Wang et al., 2020).

The local processes acting at the mesoscale and smaller scales not only alter the local hydrography and circulation, but also feed back to the large scale. State-of-the-art climate and earth system models would benefit from advancing our knowledge of these processes and improving parameterization (Hewitt et al., 2020).





Although present-day field campaigns may be able to observe mesoscale currents and features by remote sensing at lower latitudes (Gaube et al., 2015) and as part of carefully-planned synoptic surveys (Huang et al., 2018), it is practically challenging to observe and monitor submesoscale processes owing to their ephemeral life cycle and short length scale. In the Arctic basin, mesoscale scales shrink to an order of 5 to 20 km (Nurser and Bacon, 2014; D'Asaro, 1988; Manley and Hunkins, 1985) from those of 50 to 100 km at mid-latitudes (e.g. Smith, 2007); the submesoscale correspondingly is even smaller (e.g. Timmermans et al., 2012). Observations of the mesoscale and submesoscale by remote sensing are limited in the central Arctic by seasonal and prerennial ice cover, though plans for a dedicated satellite missions exist (e.g. Gommenginger et al., 2019).

Ship-based surveys in the central Arctic Ocean can give a good impression of the large-scale state and interannual changes. However, observations of mesoscale processes remain a challenge due to the scales involved and the synopticity of ship-based vertical profiling during hydrographic surveys. In addition, they are limited by logistical constraints and high cost of operation in the pack ice to the melting season and early freeze-up, from about June to September. Further, towed or autonomous platforms, such as AUVs or gliders, remain difficult or impossible to operate in the central Arctic ice pack. Similar constraints apply to seafloor moorings (e.g. Zhao et al., 2016).

One way to overcome these constraints are ice-tethered drifting ocean observing platforms. During the past three decades, these have increasingly facilitated in situ observations to augment ship-based surveys. Examples include the Polar Ocean Profiling System (POPS, Kikuchi et al., 2007), the Ice-tethered Profiler (ITP, Toole et al., 2006), and the Ice–Atmosphere–Ocean Observing System (IAOOS) platforms (Koenig et al., 2016; Athanase et al., 2019). Providing year-round timeseries in the Transpolar Drift and the Canada Basins, they are usually aimed at large-scale monitoring, with efforts such as the Marginal Ice Zone programme in the Canada Basin (Lee et al., 2017) being the exception. Coordinated efforts particularly aim at vertically connecting the processes in the realms of the atmosphere, sea ice and ocean. However, most deployments are not aimed at resolving the mesoscale and submesoscale in the horizontal.

Several systems have contained components that were designed to measure temperature and salinity at high frequency to capture short-term fluctuations, such as those associated with submesoscale, mesoscale and internal wave variability. Examples are the WHOI ITP (Toole et al., 2006), the JAMSTEC-Compact Arctic Drifter buoys (J-CAD Hatakeyama et al., 2012), as well as the UpTempO (Steele, 2017) and WARM buoys (Hill et al., 2019) .

The details of mesoscale and submesoscale processes and their effect on water mass formation, local circulation and fluxes need to be known to understand changes on an Arctic-wide scale; in particular, with respect to the feedback of the ocean with ice and atmosphere (e.g. Zhao et al., 2016). Current lack of knowledge and incorrect representation of vertical mixing processes in climate models result in large biases in Arctic Ocean temperature and salinity (Ilicak et al., 2016) and limit our knowledge of their effect on large-scale circulation (Timmermans and Marshall, 2020). The magnitude of the mesoscale and submesoscale (e.g. Nurser and Bacon, 2014; Zhao et al., 2014; Timmermans et al., 2012) indicate the potential to detect associated ocean features with novel, autonomous in situ platforms, given the quasi-synoptic character of observations from drifting sea-ice.

Here we present a unique observational dataset from the central Arctic Ocean obtained as part of the physical oceanography work program during the Multidisciplinary Observatory for the Study of Arctic Climate (MOSAiC) experiment in 2019/20 (Rabe et al., 2022).





We designed and deployed an autonomous buoy array equipped with a number of oceanographic sensors upstream the Transpolar Drift, a significantly understudied albeit crucial region of the world ocean, to resolve processes on the (sub)mesoscale over a full seasonal cycle. Below we outline the concept, realization, data processing and -validation, as well as discuss environmental, technical and analytical challenges. We present exemplary results to showcase the potential for analysis of mesoscale and submesoscale variability, and conclude with an outlook on the wider scope of this unique dataset.

## 2  Material and methods

### 2.1  General concept

In order to address some of the knowledge gaps outlined above, a set of eight buoys carrying a suite of oceanographic sensors was deployed upstream the Transpolar Drift in October 2019/20 as part of the MOSAiC Distributed Network (DN). The instruments were installed within ∼40 km of the German research icebreaker RV *Polarstern*, which was planned to stay anchored to the same ice floe for an entire year and constituted the main MOSAiC ice camp (also referred to as "Central Observatory"). Incorporating a diverse collection of autonomous instrumentation, the DN was intended to spatially extend and complement the extensive work program conducted by the different teams (Nicolaus et al., 2022; Shupe et al., 2022; Rabe et al., 2022) in the Central Observatory, and to bridge the gap between the spot measurements and larger-scale airborne and satellite observations. The DN consisted of several complex instrument systems distributed to a dozen main sites, complemented by a large number of additional simple GPS buoys in the wider DN. Eight of these sites were equipped with the buoys presented in this study, each co-deployed with a handful of other instruments (see below). This distribution, together with the ice relative drift across the liquid ocean, covered the mesoscale (of the order of few tens of kilometres), and to some extent the submesoscale (of the order of 1 km). While we were originally aiming for less spacing between the individual buoys to better resolve the submesoscale processes, the challenging ice conditions and absence of sufficiently stable floes required us to change the plan and to extend the deployment area to reduce the risk of early failures. The exceptionally dynamic ice pack in winter 2019/20 led to a failure of a number of other buoys already in the early period of MOSAiC (including 3 of 8 units described here, see 2.3).

Our focus was on the upper water column, in particular the whole upper ocean mixed layer across most of the Eurasian Basin. To capture conditions across the Transpolar Drift, the measurements ensued away from the inflow of warm Atlantic Water through the Fram Strait and as close as possible to the Siberian continental slope. The upper 100 m were deemed sufficient to achieve that aim and to also cover part of the upper halocline throughout much of the year.

We required a sufficient number of measurement points in the vertical to give a good indication of the structure of the mixed layer and the upper halocline. This especially aimed at resolving passing eddies or fronts, whilst minimising the number of sensor packages and associated cost. Deployment alongside ocean profilers at different sites within the DN, such as the Woods Hole Ice-Tethered Profiler (Toole et al., 2006) or the Drift-Towing Ocean Profiler (Ocean University of China, 2021) ensured that the vertical structure of the upper water column was measured to 1 dbar resolution at least once a day. At the same time,





our systems aimed at capturing transient phenomena, such as internal waves or eddies, requiring measurements as frequent as two minutes.

## 2.2 Instrument description

The sensor platform, hereafter referred to as "Salinity-Ice-Tether" (SIT), is a floatable buoy built by Pacific Gyre (California, USA) and comprised an egg-shaped surface float that housed the main electronics and batteries, and a 100 m long inductive mo-

dem cable with a 10 kg terminal weight attached to the bottom (Fig. 1a). Along the tether, five SBE37-IMP MicroCAT CTDs (SeaBird Scientific, California, USA; hereafter referred to as "Seabird"). The instruments, hereafter referred to as "CTDs" (Conductivity–Temperature–Depth), were mounted at depths of 10, 20, 50, 75 and 100 m. In this paper, we refer to the individual CTDs by the buoy ID and nominal CTD deployment depth, i.e. 2019O1-10m is the CTD attached to the tether of buoy 2019O1 at a nominal depth of 10 m.

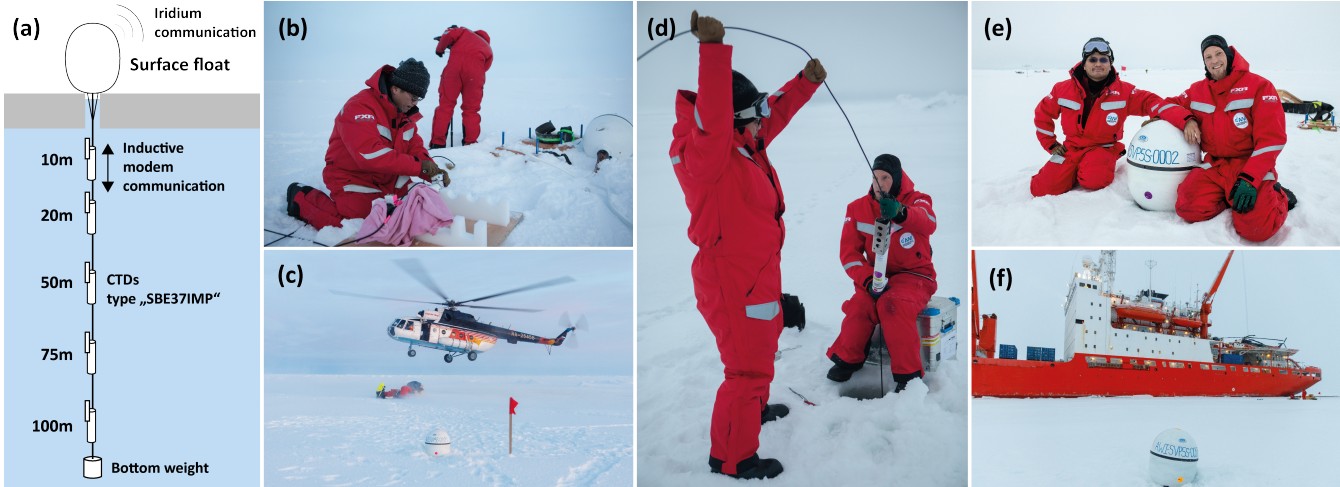

**Figure 1.** Salinity Ice Tether buoy schematic (a) and deployment photos (b-f).

The surface buoy recorded GPS position and surface temperature, and additionally carried a submergence sensor. The individual CTDs were set to record internally at 2 minute intervals, and were polled for an additional measurement by the buoy itself according to its measurement interval. The buoy reported GPS and CTD data at a default interval of 10 minutes via the iridium network.

  Upon removal of a magnet at the side of the hull after deployment, the buoy was switched on and automatically activated the

internal 2 minute sampling of the attached CTDs. The buoy controller sampled all of the CTDs one minute before a scheduled transmit, independent of the CTD's automatic sampling. The buoy controller sampled the GPS at the scheduled transmit time in order to get a valid GPS position for transmission. All data were then compressed into a file and sent via the Iridium short burst data (sbd) protocol to a shore-based server, which decoded the data and provided the data for download. In the standard buoy firmware, pressure is by default rounded to 0.1 dbar before transmission to save data size and transmission cost. Unfortunately,





this could not be changed during operation. Salinity was calculated on the server side based on the transmitted temperature, conductivity and pressure measurements according to the Practical Salinity Scale 1978. For the final dataset however, practical salinity was recalculated from the other variables using the Matlab GSW toolbox (McDougall and Barker, 2011).

The SBE37 MicroCAT CTD itself (either in its normal or inductive modem configuration) is a reliable and accurate sensor that has been used for studies of physical oceanography on various platforms all over the world for decades; in particular, for
moored operations (e.g. Beszczynska-Möller et al., 2012). The stated initial accuracy for this sensor type is $\pm0.003$ mS cm$^{-1}$ for conductivity, $\pm0.002$ °C for temperature and $\pm0.1$ % of the full scale range for pressure. For our instruments, the pressure rating was between 100 and 1000 m due to the limited availability of 100 m pressure sensors at the time of manufacturing. The initial accuracy for a 100 m calibrated pressure sensor (0.1 % of the full scale range) is thereby ∼0.1 dbar. As stated earlier, the buoy firmware only transmitted pressure records to the first decimal, resulting in a vertical resolution of 0.1 dbar
compared to 0.02 to 0.002 dbar (0.002 % full scale range) of the CTDs themselves. Since the accuracy is on the same order of magnitude, we do not consider the transmission limitation for pressure as significant in relation to our final dataset. The typical sensor stability is rated as 0.003 mS cm$^{-1}$ and 0.0002 °C per month for conductivity and temperature, respectively, and 0.05 % of the full scale range (pressure) per year.

## 2.3 Field deployment and recovery

The buoys were deployed along with a suite of other platforms during the setup phase of the MOSAiC Distributed Network by the Russian icebreaker *Akademik Fedorov* in early October 2019 (Fig. 1b-f). A few days prior to reaching the deployment destination, the instruments were successfully tested on deck. Ice floes were selected based on inspection of high-resolution satellite imagery and helicopter surveys. The ice conditions were challenging for permanent installations since thin ice was predominant in the deployment area (Krumpen et al., 2020). The station work was either done from the ship or by MI-8
helicopters. The instruments were co-installed on selected ice floes together with SIMBA-type ice mass balance buoys (Jackson et al., 2013), Snow Buoys (Nicolaus et al., 2021a) and D-TOP Ocean Profilers.

The Snow Buoy, SIMBA and SIT buoys were installed close to each other, while the D-TOP was installed at a distance of at least 70 m away. The buoy tether was laid out on the ice, and the CTDs were attached to their designated depth positions along the cable. Air temperatures were as low as -15 °C , and the instruments were covered with towels for protection. After
a deployment site with sufficiently stable level ice was chosen, the cable and instruments were manually lowered into the ocean through a 10 inch hydrohole, and the surface float was placed on top of the hole. Table 1 summarizes the deployment information for each instrument.

The trajectories of the buoys during their 10 months long drift are presented in section 4.2. 2019O2, 2019O7 and 2019O8 failed within the first 5 weeks after their deployment. 2019O5 failed in July 2020, most likely due to ice ridging, having
survived for 9 months. The other four buoys were recovered from the decaying ice field in the marginal ice zone of Fram Strait in August 2020 by the German and Russian icebreakers RV *Polarstern* and *Akademik Tryoshnikov*.





**Table 1.** Buoy deployment and recovery metadata.

| Buoy ID | Site | Deployment Date | Lon | Lat | Last data | Lon | Lat | Days |
|---------|------|-----------------|------|------|-----------|------|------|------|
| 2019O1 | M1 | 20191005T05:10 | 131.27 | 84.92 | 20200805T10:00 | -1.54 | 78.59 | 305 |
| 2019O2 | M2 | 20191007T02:40 | 135.76 | 84.87 | 20191113T22:50 | 120.14 | 85.95 | 40 |
| 2019O3 | M3 | 20191007T07:20 | 137.83 | 85.05 | 20200803T12:20 | -4.35 | 78.63 | 301 |
| 2019O4 | M4 | 20191008T01:30 | 136.28 | 85.11 | 20200814T07:10 | -8.53 | 80.13 | 311 |
| 2019O5 | M5 | 20191009T02:50 | 139.05 | 85.05 | 20200712T15:50 | -1.69 | 81.32 | 278 |
| 2019O6 | M6 | 20191010T03:20 | 133.23 | 85.13 | 20200813T13:40 | -7.49 | 80.25 | 308 |
| 2019O7 | M7 | 20191011T02:40 | 135.84 | 84.74 | 20191025T20:20 | 128.36 | 85.32 | 15 |
| 2019O8 | M8 | 20191011T01:30 | 134.50 | 84.99 | 20191025T20:50 | 126.32 | 85.55 | 15 |

## 3 Data processing & quality flags

The final data collection consists of eight individual datasets, one for each buoy. Four are merged products of the initially iridium-transmitted buoy data (10 minute interval) and the corresponding CTD data from recovered instruments (2 minute interval). In order to ensure the highest possible data quality and at the same time leave the end user the possibility to apply his/her own quality checks, we decided to not entirely remove questionable data, but to instead apply a primary quality flagging scheme to the oceanographic data in a modified version of the scheme proposed in UNESCO-IOC (2013). The primary level flagging scheme we use here is composed of five quality values which are defined as follows: 1 — good, 2 — modified, 3 — questionable, 4 — bad, 9 — missing data. The difference to the originally proposed scheme is that "2 — not evaluated" was replaced by "2 — modified", as we think it should be general practice to evaluate all data points. In some cases, as also described below, the quality of the data could be improved by modification of the original record, using the "2 — modified" flag. We applied this scheme to the oceanographic data only. We decided against using an additional, secondary flagging scheme, as further discrimination of the quality checks described below would not provide any substantial added value to our data.

Each individual dataset was compiled in the following way: First, the buoy data was obtained from the Pacific Gyre web server. A minor number of obvious GPS outliers were removed. Buoy drift speed was calculated from the difference between adjacent GPS positions, and an upper threshold of 0.8 m s$^{-1}$ was applied. Surface temperature data quality was controlled for plausibility, and no correction was necessary. The polled sampling of the CTDs had been initiated by the buoy one minute before acquisition of GPS position and time. For consistency and simplicity, we used the time of CTD sampling as the general timestamp (rather then the GPS fix one minute later), accepting an additional uncertainty in the GPS position of <30 m, depending on the drift speed. If higher temporal resolution CTD data was available for a particular buoy, both datasets were merged (see below). No quality flags for position or surface temperature are given, since their quality is overall good and the oceanographic measurements are the focus of this observational dataset. Data from the 20 recovered CTDs (corresponding





buoy IDs: 2019O1, 2019O3, 2019O4 and 2019O6) were downloaded using the Seaterm V2 software by Seabird. For logistical reasons, data from 2019O1 and 2019O3 were downloaded at sea as .cap files, whereas data from 2019O4 and 2019O6 were
recovered at home after the end of MOSAiC as standard .hex, .xml and .xmlcon files and converted to .cnv files using the Seabird software SBEDataProcessing.

The CTD timestamps were checked for consistency using distinct features in the pressure records of each set of CTDs installed on any given buoy (for example when all instruments would be lifted up at the same time due to an increased drift speed as a result of strong winds). 2019O4-75m was mistakenly configured to a 30 s measurement interval, which led to an
early power failure (see below). For this CTD, only every fourth record was used for the final dataset for consistency reasons. The full dataset is however still available in the raw dataset. Also, the instrument time of 2019O4-75m had an offset of 56s, which was corrected.

For the four merged data products, the data were combined and reordered based on their timestamps. All records prior to the deployment have been removed from the dataset (not flagged) based on the pressure record of the instruments. In some
instances, the initial conductivity values appeared suspiciously low, sometimes even with sudden jumps to higher values. There is a high probability, though no complete certainty, that these are erroneous records were caused by ice formation in the pump or conductivity cell that resulted from the instrument being exposed to cold air temperatures prior to deployment through the ice. In most instances, this effect disappeared after a few days to several weeks. We tried to identify a point in the time series where the salinity data starts to become reasonable; for example, after the last suspicious jump, by plausibility checks and
comparison against adjacent CTDs, or comparison to other buoy data. After a specific point in time was identified from where the data seemed plausible, we flagged all prior records as "3 — questionable". Temperature and pressure records were not affected by this. While the latter were generally clean and reliable as expected from these sensors, conductivity measurements sometimes also exhibited suspicious values that were potentially caused by particles in the cells. Conductivity values below 20 mS cm$^{-1}$ and above 40 mS cm$^{-1}$ were flagged as "4 — bad"; at time, this coincided with the conductivity issue outlined
above. For the five longest timeseries ($sim$10 months), a moving average filter was tested and tuned to identify the outliers that still fell within the allowed global range. No window size was found that would exclusively catch the outliers, so we decided to rather flag the remaining suspicious records manually, as "3 — questionable". Further, 2019O3-10m exhibited a small but suspicious, several-day long conductivity drop in June 2020, which we subsequently flagged as "3 — questionable". Buoy 2019O6 exhibited more outliers than the other buoy datasets. Since these outliers were not found on the recovered CTDs
themselves, we conclude that this was caused by an issue in the inductive modem communication, rather than being a problem of the CTD sensor measurement. Since these outliers were too numerous for manual flagging, a moving average filter (window size 14) was applied to determine and flag these records as "3 — questionable" (~1000 total records for 2019O6). The few remaining outliers (<20) were manually flagged. Interestingly, in some instances this inductive modem issue resulted in records being wrongly assigned; i.e., a measurement taken by 2019O6-10m was assigned by the buoy to 2019o6-20m. In these cases,
the records were reassigned and flagged as "2 — modified". 2019O6 temperature and pressure suffered the same problem. Temperature was checked using subsequent moving average filters (window sizes 14 and 11; ~1900 outliers flagged as "3 — questionable"), and the remaining few outliers were manually flagged as either "3 — questionable" or "2 — modified".





When CTD data was available, the 0.1 dbar resolution buoy pressure records were linearly interpolated using the CTD pressure records in order to achieve a consistent pressure timeseries. The interpolated records were flagged as "2 — modified"

accordingly. For the periods where CTD data was available, this procedure also removed the pressure outliers caused by the inductive modem issue from the 2019O6 record, so no additional filtering was necessary. Each CTD record for a given buoy was assigned a GPS position by linear interpolation of the corresponding buoy GPS record, and drift speed was recalculated accordingly. Instrument depth was calculated from clean pressure readings and latitude using the Matlab GSW toolbox function "gsw_z_from_p", (McDougall and Barker, 2011).

Although salinity was initially calculated and provided on the server side, we recalculated this variable from conductivity, temperature and pressure according to the PSS-78 algorithm (Matlab GSW tolbox function "gsw_SP_from_C"). Quality flags for calculated depths and salinities were inherited from T, C and p. Finally, salinity was despiked using a moving average filter with a window size of 12, (equivalent to between 20 and 120 min) depending if the dataset contained 2 min or 10 min data. The final (<120) outliers identified by this filter were flagged as "3 - questionable". An overview of the processing and flagging

procedure is given in Fig. 2.

Post-deployment calibration was performed at Seabird for three sets of CTDs (2019O3, 2019O4, 2019O6; 15 in total) in July 2021. 2019O1 was re-deployed during the final MOSAiC leg a few weeks later as 2020O10 (not shown here), so the five corresponding CTDs were not available for post-deployment calibration. However, we decided to not apply any correction on the temperature and salinity data based on this calibration information (see below).

## 4   Results

### 4.1   Dataset description

The processing described above yielded eight individual time series, one for each buoy. Three time series only comprise a few weeks, while five of them are ~10 months long. For 2019O1, 2019O3, 2019O4 and 2019O6, we supply a merged product that combines buoy and CTD data.

The measured oceanographic variables are conductivity, temperature and pressure, the derived variables are salinity and depth. A primary quality flag (1 — good, 2 — modified, 3 — questionable, 4 — bad, 9 — missing data) is given for each of these 5 variables. Each measurement has a corresponding timestamp. Only the buoy measurements (indicated by a "buoy_flag") initially had a GPS record, and a position was given to the higher resolving CTD records by linear interpolation. The drift speed was calculated from the difference between GPS positions. Additionally, the buoy measured surface temperature and a

"submerged boolean", which indicates whether the buoy was in water or not. Refer to Table 2 for a summary of all variables in each processed dataset. An overview of all individual datasets and the corresponding collections for raw and processed data is given in Table 3.

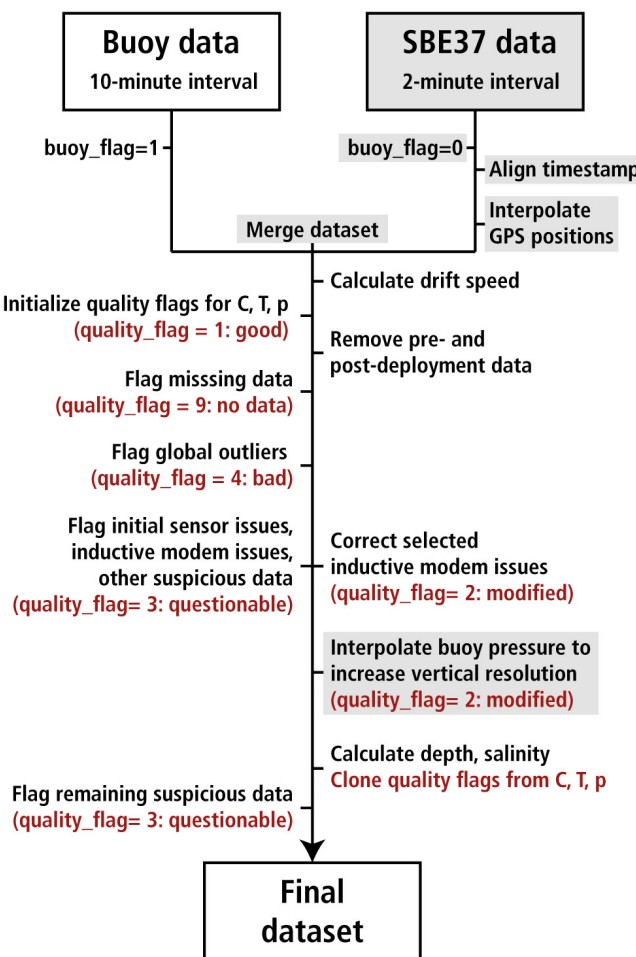

**Figure 2.** Schematic outlining the individual steps of data processing and quality control using primary flags (indicated in red) according to a slightly modified version of UNESCO-IOC (2013). The additional processing steps required when merging buoy and recovered CTD data are indicated in grey. Procedures to the left of the processing arrow represent flagging without modification of the original data, whereas processing steps to the right involve removal and modification of original data, or calculation of secondary parameters.



**Table 2.** Variables of the processed dataset. The nominal depths of the 5 CTDs (represented by Gear ID 1-5) were 10m, 20m, 50m, 75m and 100m. The actual depths are given by the "Depth" variable.

| Variable | Unit/Format | Accuracy |
|---|:---:|:---:|
| Date & Time | ISO-8601 | 1 min |
| Latitude | deg | |
| Longitude | deg | 30 m |
| Drift speed | m/s | 0.1 |
| Submerged Boolean | logical | |
| Temperature, surface | °C | 0.01 |
| Gear ID | 1 - 5 | |
| Conductivity, seawater | ms/cm | 0.003 |
| Temperature, seawater | °C | 0.002 |
| Pressure, seawater | dbar | 0.1 |
| Depth, instrument | m | 0.1 |
| Salinity, seawater | PSU (PSS-78) | 0.01 |
| buoy_flag | logical | |
| C_flag | 1 - good | |
| T_flag | 2 - modified | |
| p_flag | 3 - questionable | |
| D_flag | 4 - bad | |
| S_flag | 9 - missing data | |





**Table 3.** Datasets overview (*links subject to change after DOI registration in accepted version)

| Level | Buoy ID | Reference | Link |
|---|---|---|---|
| Raw data | 2019O1 | Hoppmann et al. (2021a) | https://doi.pangaea.de/10.1594/PANGAEA.933934 |
| | 2019O2 | Hoppmann et al. (2021b) | https://doi.pangaea.de/10.1594/PANGAEA.933928 |
| | 2019O3 | Hoppmann et al. (2021c) | https://doi.pangaea.de/10.1594/PANGAEA.933932 |
| | 2019O4 | Hoppmann et al. (2021d) | https://doi.pangaea.de/10.1594/PANGAEA.933933 |
| | 2019O5 | Hoppmann et al. (2021e) | https://doi.pangaea.de/10.1594/PANGAEA.933937 |
| | 2019O6 | Hoppmann et al. (2021f) | https://doi.pangaea.de/10.1594/PANGAEA.933941 |
| | 2019O7 | Hoppmann et al. (2021g) | https://doi.pangaea.de/10.1594/PANGAEA.933939 |
| | 2019O8 | Hoppmann et al. (2021h) | https://doi.pangaea.de/10.1594/PANGAEA.933942 |
| Collection | all | Hoppmann et al. (2021i) | https://doi.pangaea.de/10.1594/PANGAEA.937271 |
| Processed data | 2019O1 | Hoppmann et al. (2022a) | https://doi.pangaea.de/10.1594/PANGAEA.940271 |
| | 2019O2 | Hoppmann et al. (2022b) | https://doi.pangaea.de/10.1594/PANGAEA.940298 |
| | 2019O3 | Hoppmann et al. (2022c) | https://doi.pangaea.de/10.1594/PANGAEA.940282 |
| | 2019O4 | Hoppmann et al. (2022d) | https://doi.pangaea.de/10.1594/PANGAEA.940291 |
| | 2019O5 | Hoppmann et al. (2022e) | https://doi.pangaea.de/10.1594/PANGAEA.940301 |
| | 2019O6 | Hoppmann et al. (2022f) | https://doi.pangaea.de/10.1594/PANGAEA.940296 |
| | 2019O7 | Hoppmann et al. (2022g) | https://doi.pangaea.de/10.1594/PANGAEA.940303 |
| | 2019O8 | Hoppmann et al. (2022h) | https://doi.pangaea.de/10.1594/PANGAEA.940304 |
| Collection | all | Hoppmann et al. (2022i) | https://doi.pangaea.de/10.1594/PANGAEA.940320 |

## 4.2 Drift trajectories

The drift trajectories of all eight buoys are shown in Figure 3. Their journey alongside the Central Observatory, from the
deployment area north of the Laptev Sea through the Transpolar Drift until their recovery in Fram Strait, took about 10 months.
The buoys travelled over the Gakkel Ridge from the Amundsen Basin to the Nansen Basin in March/April 2020.

The drift accelerated strongly from June onwards upon reaching the Yermak Plateau. While a detailed discussion of the drift
pattern is beyond the scope of this paper, it should be noted that the relative positions of the buoys within the array did not
change much before finally entering the Fram Strait area (Figure 3).

## 4.3 Oceanographic data

In the following, we take a closer look at the individual processed datasets (Figs. 4, 5).

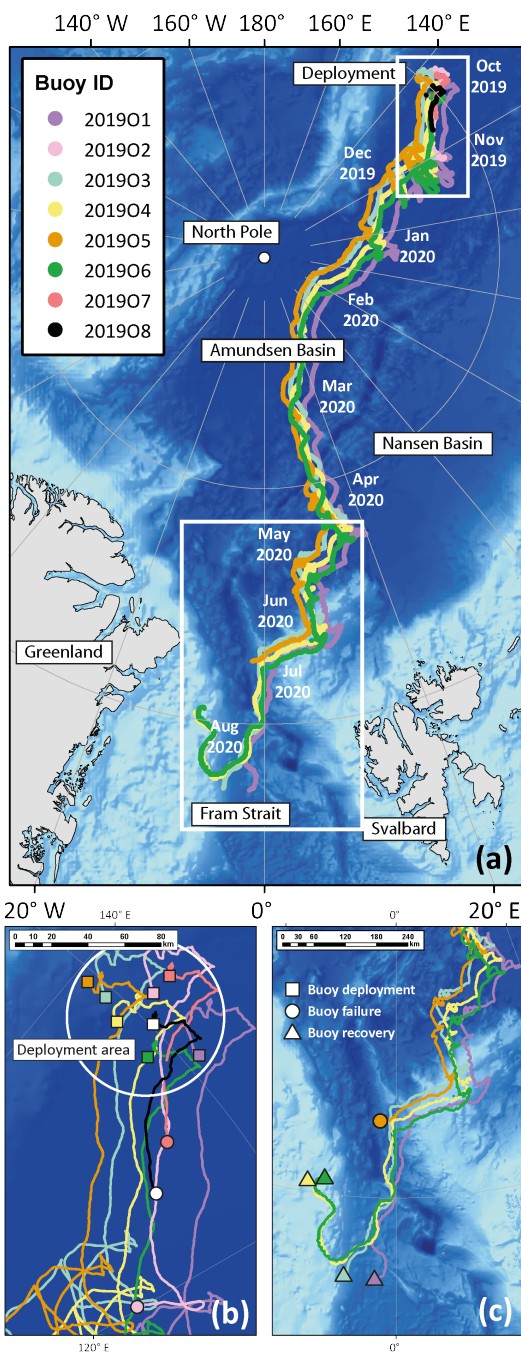

**Figure 3.** Maps of buoy drift trajectories through the Arctic Transpolar Drift between October 2019 and August 2020. a) Overview map; b) initial setup north of Laptev Sea and first weeks of drift; c) last weeks of drift in Fram Strait. Squares indicate deployment locations, circles indicate locations of failure or recovery. Bathymetric data was obtained from the IBCAO project (Jakobsson et al., 2020). Coastline data is taken from Wessel and Smith (1996).

As stated earlier, we refer to a specific instrument time series as "buoy name - nominal instrument depth". For simplicity, we use the buoy name and instrument references as a synonym for the time series itself.

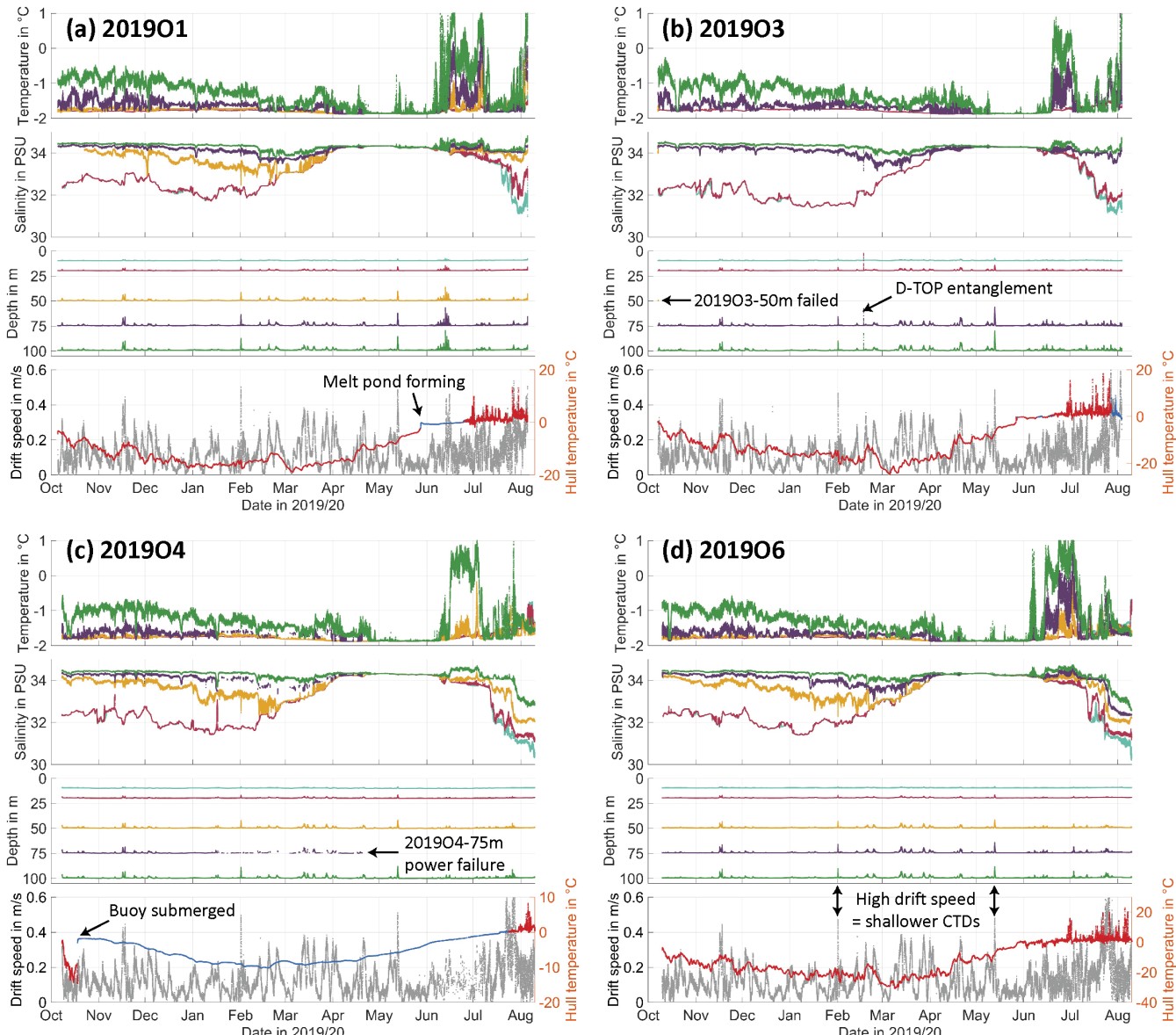

**Figure 4.** Time series of selected variables recorded by buoys and CTDs a) 2019O1; b) 2019O3; c) 2019O4 and d) 2019O6. The panels each show from top to bottom: seawater temperature (ITS-90), seawater practical salinity (PSS-78), instrument depth, buoy drift speed (grey) and surface temperature (red indicates submerged in water, blue indicates not submerged).

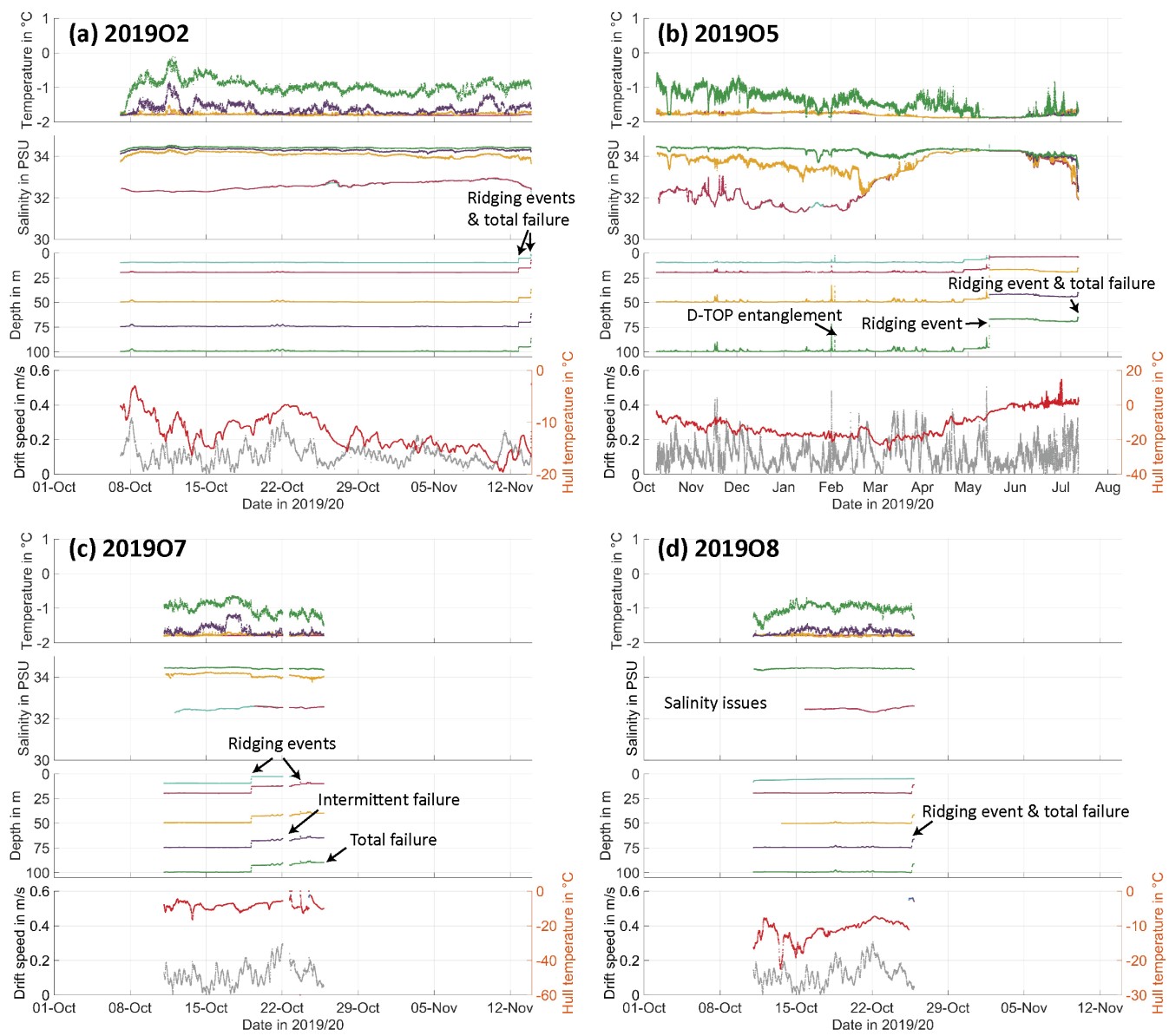

**Figure 5.** Time series of selected variables recorded by buoys (no recovered CTD data) a) 2019O2; b) 2019O5; c) 2019O7 and d) 2019O8. The panels each show from top to bottom: seawater temperature (ITS-90), seawater practical salinity (PSS-78), instrument depth, buoy drift speed (grey) and surface temperature (red indicates submerged in water, blue indicates not submerged).





2019O1 was generally performing well, except some problems at the beginning of its life cycle. 2019O1-10m and 2019O1-20m showed a suspiciously low salinity for the first few days after deployment and then adjusted to consistent levels. 2019O1-50m exhibited a similar effect, but it took a 2.5 weeks before plausible data were collected. This was likely caused by ice in the conductivity cell. 2019O1 continued to collect reasonable data throughout its operational time. The unit was recovered on 5 August 2020. 2019O2 started out fine, before all instruments were suddenly lifted up by 5 m over the course of one hour during an event on 12 November 2019. They remained in place for one day, before they were lifted again by 8 m in a similar event. The buoy was lost immediately afterwards on 13 November 2019. 2019O3 was fine initially, but 2019O3-50m stopped working after several hours following the deployment, and never resumed transmitting. The other 4 instruments were performing well throughout their operational time. During an event on 17 February 2020, all CTDs were substantially lifted up although there was no particularly strong drift at that time. In fact, 2019O3-10m and 2019O3-20m were even lifted as shallow as the ice base, which is unusual. The only explanation is that the buoy tether got entangled with the co-deployed D-TOP free floating profiler, which possibly got moved closer to the SIT as a results of converging pack ice conditions. The sensors returned to their original depths after ∼30 min. The remaining sensors continued to measure correctly until the buoy was recovered on 03 August 2020. 2019O4 was fine from the start. Due to an unintentionally high measurement interval of 30 s, 2019O4-75m ran out of power on 20 April 2020 after a period of intermittent failures starting on 11 January 2020. The buoy and all other CTDs continued to work well until recovery on 14 August 2020.

2019O5 showed some moderate problems. 2019O5-75m stopped communicating upon deployment. 2019O5-10m started to exhibit inductive modem communication issues at the end of October 2020. This began with sporadic missing measurements in the buoy's transmitted data ("blanks"), which subsequently became more frequent throughout November, when roughly half of the measurements were blanks. In the second half of November, the issue gradually disappeared, with transmissions back to normal on 27 November. From late December 2019 onwards, 2019O5-20m and 2019O5-50m started to exhibit similar issues, again first sporadic, then almost continuously (roughly half of the measurements affected). Interestingly, this problem never affected both sensors at the same time, the missing transmissions were always "alternating", and sometimes more or less periodic. From mid January onwards, 2019O-50m recovered from this problem, and transmissions were completely normal from end of January 2020 onwards. However, the issue with 2019O5-20m persisted until mid February 2020, with failures still being somewhat periodic at times, when the blanks slowly started to decrease. In late February, the problem with 2019O5-20m was also gone, when 2019O5-100m started to exhibit blanks, with the same pattern: first sporadic, then much more frequently. In late April they became only sporadic again, with a very small fraction of blanks during the remaining buoy lifetime. The reason for this behaviour remains unclear. On 14/15 May 2020 a drastic event took place: 2019O5-50m and 2019O5-100m showed a sudden 30 m pressure decrease on that day, suggesting that the entire buoy was lifted up by that amount, presumably in a substantial ridging event. Communication to 2019O5-10m was lost in the process, as it was probably torn off the tether. At the same time however, 2019O5-75m started to transmit data again, at a current depth of ∼45 m (as expected as the event lifted the instrument from a nominal depth of 75 m). 2019O5-20m was only lifted up by 16 m, to a depth of 4 m, suggesting that it was violently pushed down along the tether by the resistance of the ice bottom. Surprisingly, the sensor survived this incident. The buoy kept collecting reasonable data via the four remaining CTDs (2019O5-20m presumably directly from the



ice base), and interestingly without any further inductive modem issues. There were two more events, on 13 and 16 June 2020,
during which three of the four instruments suddenly moved down 1 m each, interestingly with the exception of 2019O5-20m,
which was then at 3.8 m and presumably somehow stuck in the ice. Another event during which the buoy was lifted up by 4
m occurred on 12 July 2020, briefly before the buoy stopped transmitting any more data. 2019O6 was generally performing
well, but suffered from an issue with the inductive modem link between the buoy and the CTDs. 2019O6-10m had some minor
conductivity cell issues in the first few days after deployment. The buoy started to suffer from inductive modem communication
issues with all instruments on the tether, which became worse from November 2019 onwards, but significantly improved again
from 13 April 2020 for unknown reasons. These issues resulted in erroneous values being recorded by the buoy (in contrast to
the problems with 2019O5, where measurements were just blank), leading to a large number of outliers in the raw dataset. On
03 August 2020, all CTDs exhibited a sudden drop in pressure of 0.5 dbar, probably related to a draining meltpond (although
the submergence sensor was never triggered) or a similar effect which caused the surface buoy to slightly drop. The unit was
recovered on 13 August 2020. 2019O7-75m suffered from conductivity cell issues upon deployment and did not recover. All
instruments were lifted up by 7 m during an event on 19 October 2019. The instruments were at stable depths for a few days,
before they were lifted again over a period of a few days between 22 and 24 October. 2019O7-10m was lost eventually, and the
buoy stopped transmitting entirely on 26 October 2019. Three instruments on 2019O8 had issues with ice in the conductivity
cell upon deployment. On 25 October 2019, all instruments were lifted up by  10m over the course of a few hours, with
2019O8-10m being stuck at the ice bottom and pushed down the tether. A few hours after that, the buoy stopped transmitting.

## 5  Discussion

In this section, we assess the quality of the oceanographic data and discuss the general instrument performance. We showcase
the strength of the deployment concept and potential of the dataset as a whole, and assess the wider role of the data, in particular,
the potential within in framework of MOSAiC.

### 5.1  Data quality & validation

In addition to a general data plausibility and consistency check among individual (independent) CTDs installed on the same
tether as performed during the data processing, there are a few other methods to assess the general data reliability and quality.
These will be elaborated on in this section.

#### 5.1.1  Post-deployment calibration

The CTDs were manufactured and laboratory-calibrated by Seabird only 2 months before deployment. Unfortunately, due to
logistical reasons no dedicated in situ calibration cast by the attachment of the individual CTDs to a higher-accuracy ship-
based system could be performed, neither pre deployment nor post deployment. However, there was the possibility of a cross-
validation to observations taken nearby during visits to the sites (see below).





**Table 4.** Temperature offset and conductivity drift from post-deployment calibration. Pressure (not shown) did not drift at all. This information is provided here for reference. Corrections have not been applied to the data (see text).

| Buoy | Depth | SN | pre-cal | post-cal | C-slope | T-offset in mdeg |
|---|---|---|---|---|---|---|
| 2019O3 | 10 | 21100 | 18-Aug-2019 | 13-Jul-2021 | 0.9999781 | 0.44 |
| | 20 | 21103 | 18-Aug-2019 | 09-Jul-2021 | 1.0002158 | -0.06 |
| | 50 | 21113 | | mainboard broken | | |
| | 75 | 21114 | 16-Aug-2019 | 08-Jul-2021 | 0.9996651 | 0.57 |
| | 100 | 21084 | 12-Aug-2019 | 15-Jul-2021 | 0.9999438 | -0.19 |
| 2019O4 | 10 | 21093 | 18-Aug-2019 | 13-Jul-2021 | 0.9995212 | 0.35 |
| | 20 | 21094 | 18-Aug-2019 | 09-Jul-2021 | 0.9999413 | 0.07 |
| | 50 | 21095 | 14-Aug-2019 | 13-Jul-2021 | 1.000185 | 0.39 |
| | 75 | 21099 | 25-Aug-2019 | 13-Jul-2021 | 1.0002775 | 0.6 |
| | 100 | 21083 | 12-Aug-2019 | 27-Jul-2021 | 0.999825 | 0.15 |
| 2019O6 | 10 | 21112 | 19-Aug-2019 | 08-Jul-2021 | 0.9999426 | 0.29 |
| | 20 | 21115 | 18-Aug-2019 | 13-Jul-2021 | 1.0001306 | 0.21 |
| | 50 | 21116 | 18-Aug-2019 | 13-Jul-2021 | 1.0001337 | 0.4 |
| | 75 | 21117 | 18-Aug-2019 | 13-Jul-2021 | 1.0000979 | 0.69 |
| | 100 | 21108 | 19-Aug-2019 | 13-Jul-2021 | 1.0005926 | -0.69 |

The results of the post deployment calibration were within normal limits for this instrument type (Table 4). While a tem-

340 perature offset and conductivity drift correction of the data according to Seabird's Application Note 31 was considered, we decided not to apply this procedure to the present dataset for the following reasons: the offset in temperature (usually mainly by electronic drift) was generally on the order of $\pm0.0005$ °C , which is within the noise level of the instruments. In addition, the calibration sheets show that the main contribution to the calculated offset stemmed from bath temperatures $>5$ °C , which is much higher than temperatures measured by our instruments in the field. The conductivity drift, which is usually mainly a result

345 of fouling processes within the conductivity cell, was also generally low. The problem with fouling is that the usual assumption of a linear change is not necessarily true. Such a change can also occur suddenly, and the calibration does not account for the timing. Assuming the fouling would lead to a linear drift of the cell, the instrument is treated in various ways and exposed to different environmental conditions between the recovery and the actual calibration. It is usually cleaned, packed, transported and stored, so the calibration results are not necessarily reflecting the conditions under which the instrument measured. Finally,

350 an open question is whether to perform the linear correction between the calibration dates (as recommended by Seabird), or





between deployment and recovery, where fouling can actually occur. Taking all these issues into account, we considered the calibration results more as a validation whether an instrument is generally reliable or not, and leave the correction to the end user. The results from our calibration procedure suggest that all instruments in Table 4 can be considered reliable. As a final note, 2019O3-50m was diagnosed with a mainboard failure and was replaced on warranty.

### 5.1.2 Validation using independent measurements

A wealth of hydrographic data was collected during MOSAiC at different times and places as part of the physical oceanography program. Specifically intended to validate the buoy data in situ, several deployment sites were revisited during April and May 2020. A number of water column profiles were obtained using two handheld unpumped, self-recording CTDs (SST48M, Sea&Sun Technologies, Germany; hereafter referred to as "SST-CTD"). The instrument was mounted on the line of a fishing rod with a battery-powered winch to measure profiles of temperature, salinity and pressure in the upper water column. A total of 5 SST-CTD profiles were taken at different deployment sites (see table A1).

In addition to the manufacturer's calibration, the two SST-CTDs were mounted on a standard ship-based high-accuracy CTD-Rosette system for in situ calibration (Seabird SBE911+, hereafter referred to as "ship CTD" for simplicity) to validate the sensors themselves. These intercalibration exercises were carried out in February and July 2020, close in time to the validation casts at the buoy sites to minimize the potential influence of unaccounted sensor drift. The results of these calibration casts showed a deviation of both SST-CTD's pressure from the ship CTD of approximately 1 m at 100 m depth. This has been taken into account for the comparison to the buoy data by smoothing the profile. SST-CTD salinity and temperature deviations were largest within layers of high vertical gradients, and lowest in layers with small gradients (at 200 to 300 m depth), as can be expected for these kinds of intercalibration experiments. The average salinity and temperature offsets were 0.008 PSU and 0.006 °C , respectively, and have been corrected.

For comparison with the buoy data, the SST-CTD validation profiles have been smoothed using a Savitzky-Golay filter (window size of 31, corresponding to about 3.5 m). An example of the comparison is shown in Figure 6.

The average deviation values between SST-CTD and the buoy temperature and salinity are $\delta T$=-0.002 $\pm$0.002 and $\delta S$=-0.004$\pm$0.006, respectively, which is within the stated accuracy of the SST-CTD sensors.

In conclusion, the CTDs on the buoys show very good agreement with the intercalibrated SST-CTDs after a measurement period of 6 months.

### 5.1.3 Data transfer issues

An additional source of error that is not necessarily obvious is introduced by quirks in the inductive modem communication between the instruments and the surface package of the buoy. The reliability of the oceanographic data polled from the individual CTDs by the surface electronics via the inductive modem link can be assessed by comparison of the buoy data set to the data collected internally by the 20 individual CTDs that were recovered in August 2020. This comparison confirmed that the buoys generally operated as intended. However, in the case of 2019O6, this comparison helped to identify and remove a large number of outliers that seemingly resulted from an (unknown) issue in the inductive modem communication.



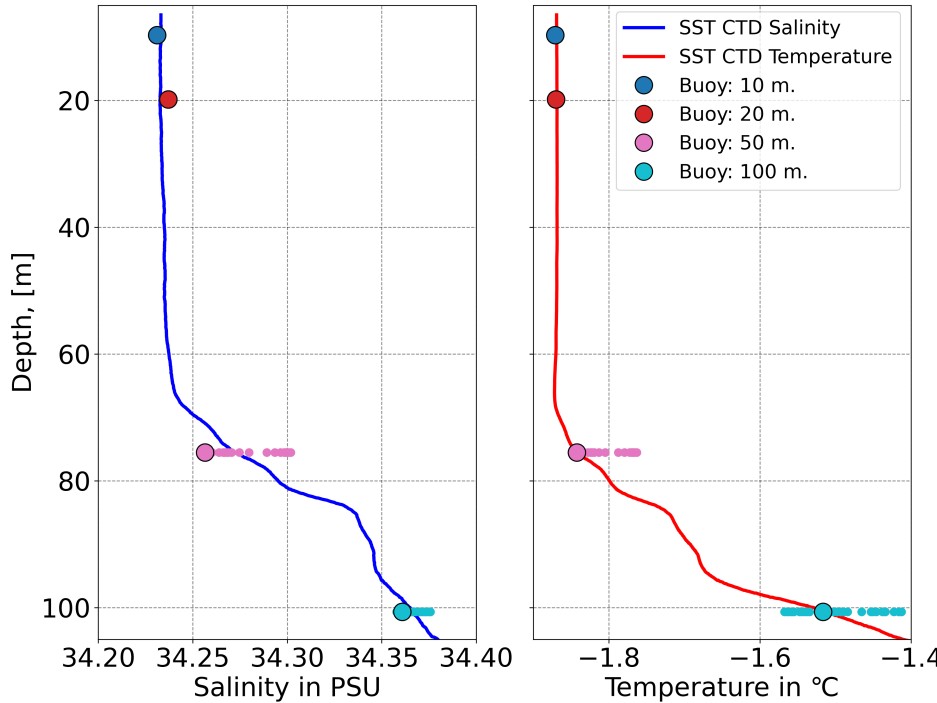

**Figure 6.** Comparison between SST-CTD and 2019O3 data on 26 April 2020. Red and blue lines represent salinity and temperature recorded by the SST-CTD, respectively. Circles represent buoy data closest in time to the SST-CTD cast. The colored dots show the values of the corresponding buoy CTD within half an hour before and after the SST-CTD cast. Buoy data at 10 and 20 m are hidden below the circles.

The transmission of data via iridium (in this case the short burst data protocol) may be sometimes unstable due to a variety of reasons, but usually only leads to the unit not being able to transfer a package in the first place, or more rarely, to the loss of individual packages. This technique generally does not introduce errors in the recorded data itself as far as we are aware.

## 5.2 Properties of an observed eddy

Here we present a prominent example of the type of features observable with the present instrumentation and overall approach: a mesoscale eddy that passed through the buoy array in February 2020.

During that time, the MOSAiC observatory drifted towards the northwest at an average velocity of about $0.13$ m s$^{-1}$. The amount of loops were considerably less compared to the previous November and December (Fig. 7a). The buoys drifted along a pronounced surface salinity gradient, from less saline water in the southeast to more saline in the northwest (Fig. 7). The lateral gradients of salinity and temperature are evident from both, the time series along each individual buoy track, as well as in the observations between different buoys.

As evident from the CTD records at 20 and 50 m, the depth of the upper ocean mixed layer (ML) increased from between 20 and 50 m in early February to >50 m later on. Changes in water properties with time at 20 and 100 m depths are shown in the



**Figure 7.** Preliminary observations of an eddy passing through the Distributed Network in February 2020. a) Trajectories and observed salinity at 10 m depth of 5 buoys (colored) along with trajectory of RV *Polarstern* (black) between 02 and 20 February 2020. Triangles indicate the buoy positions on 02 February. The colored horizontal lines indicate start and end positions of the shaded areas in c and d, where changing water properties were observed by the corresponding buoy. b) T/S diagram of buoy data at 20 and 100 m depth during 01 to 20 February 2020 (red) and 20 February to 01 March 2020 (blue). c) and d) Temperature and salinity recorded at four different depths by buoys 2019O1, 2019O4 and 2019O6. The line colors and shaded areas correspond to the triangle colors in a.



temperature-salinity diagram in Figure 7b). The mixed-layer salinity increased with a corresponding decrease in temperature during February. At the same time, salinity and temperature at 100 m decreased. An increase in mixed-layer salinity could be partly explained by an intensification of the vertical mixing during two storms that took place in February, with wind speeds

up to 16 ms$^{-1}$, and with ice formation at the surface. However, close-by near-daily measurements of the "vertical mixing properties" with a dedicated microstructure probe at the central observatory did not show significant mixing underneath the halocline (K. Schulz, pers. comm.).

Further mixed-layer deepening occurred into early April (Figure 4) when the ML depth reached 100 m. Between February and April, the drift path of the buoys was about 340 km, or 270 km air-line distance. They drifted across the front associated

with the transition from the Transpolar Drift to the warm and salty inflow of water of Atlantic origin through the Fram Strait. These types of upper ocean salinity gradients generally provide a high potential to form eddies.

Between 7 and 15 February, rapid changes in the water properties were registered by 2019O4, 2019O6, and, to a lesser extent, 2019O1. 2019O6 (blue triangle and lines) and 2019O4 (red triangle and lines) registered these changes four and one days in advance of Polarstern reaching the same location, respectively. This feature indicated the presence of an eddy with

approximate locations registered by the buoys as shown by two short lines (beginning and end) on the buoy tracks (Fig. 7a) and colored shaded areas in Figure 7c,d with buoy temperatures and salinities at four depths, 20, 50, 75 and 100 m. The CTDs at 10 m depth show mostly the same water properties as those at 20 m. Line colors correspond to the triangle colors of the buoys in Figure 7a. The position of the eddy remains approximately the same. This event was also observed by the current measurements obtained by the shipboard Acoustic Doppler Current Profiler (ADCP). It showed the presence of the eddy between 11 and 13

February (S. Tippenhauer, pers. comm.), and the location is denoted in Figure 7a (black dots on the dashed line). Most likely, the ship drifted close to the centre of the eddy. It exhibited a rather symmetric current structure, with the minimum speed in the center and currents on the sides in opposite directions with speeds up to 20 m$^{-1}$. The estimated diameter of the eddy was about 30 km. A detailed description of the eddy properties is beyond the scope of this paper, and is the subject of ongoing work.

It is important to note that the drift speed increased from 0.04 to 0.2 m$^{-1}$ between 07 and 14 February. Thereby, the time

interval when the eddy was observed by different buoys decreased while the drifting distance (or eddy diameter) remained similar. At the same time, 201903 and 201905 did not show significant changes at a depth of 100 m. Both buoys were located to the northeast of the ship. It is likely that they were too far away from the core of the eddy to register a measurable signal.

All three buoys that registered the eddy encountered similar anomalies in temperature and salinity. The salinity decreased by 0.22 and 0.17 at 100 and 75 m, respectively, and increased by 0.5 at 50 m depth with a corresponding temperature decrease

of 0.3, 0.12 and 0.05 °C  for buoy 201906. The buoyancy frequency, $N^2 = -\frac{g}{\rho_0}\frac{d\rho(z)}{dz}$, calculated between 50 and 100 m, decreased by factor of 2 during the passing of the eddy from $2 \cdot 10^{-4}$ to $1 \cdot 10^{-4}$ for 2019O6, from $1.6 \cdot 10^{-4}$ to $0.8 \cdot 10^{-4}$ for 201904 and from $1.1 \cdot 10^{-4}$ to $0.5 \cdot 10^{-4}$ for 201901.

The first baroclinic mode Rossby radii $L_R \equiv NH/\pi f_0$ are around 28 km, assuming $H =$80 m and $N$ calculated from data at 20 and 100 m depth, both for 2019O6 and 201904. The value for 2019O1 is about 24 km. These estimates are close to the

estimated diameter of the eddy.





The distributed nature of the buoys allowed a fully synoptic assessment at different points across the array, which could not have been achieved by just one buoy.

## 5.3 Wider scope of the dataset

The example in Section 5.2 shows how our data can be used to study a mesoscale features in the upper Arctic Ocean. The
scope of science questions that could be studied with the data is broad: the seasonality could be explored further, as well as the variability along the pathways of the TPD (e.g. Stedmon et al., 2021). In particular, the buoy data fills the usual winter gap in manual in situ observations, often carried out during seasonally limited ship surveys. The high temporal resolution allows to study passing transient phenomena, as the buoys acted as a quasi-stationary platform, depending on drift speed. On the other hand, during times of faster drift, the systems measured the the upper ocean quasi-synoptically over submesoscales. This
allows to study spatiotemporal variability of temperature and salinity. In particular, horizontal wavenumber spectra can give insights into the length scales of variability induced by sub-meso-scale restratification and internal waves (Marcinko et al., 2015; Timmermans et al., 2012). Furthermore, submesoscale eddies and filaments between mesoscale features can be studied with this piecewise quasi-synoptic dataset. The scientific results of these potential studies, as well as the data itself, could serve to validate numerical models of the ice-ocean system and the fully coupled earth system. The observations can also serve as
input to numerical models, either to initialise or force a simulation, or to provide input for assimilation. Finally, the GPS data recorded by our buoys were also used to create the "official" drift trajectories for several of the main sites of the Distributed Network (Nicolaus et al., 2021b).

## 6 Conclusions & Outlook

We have shown results from a unique deployment of a number of ice-tethered buoy systems measuring temperature and salinity
in the upper 100 m of the central Arctic Ocean. These systems were part of the MOSAiC expedition, a year-long drift of the German icebreaker RV *Polarstern* in 2019/20. The deployment concept was specifically designed to observe mesoscale and, to a lesser degree, submesoscale variability in the ocean mixed-layer and upper halocline throughout different seasons and across the Eurasian Basin. Additionally, the buoys provided valuable oceanographic data as a complement to various other co-located multidisciplinary ice-tethered buoy systems, and to the overall MOSAiC work program. The buoy systems generally performed
well, with a few expected failures due to ice deformation, in particular, as the sea ice environment was even more dynamic than anticipated. An added value was obtained by recovering four of the eight buoys, yielding data at even higher temporal-resolution. We have shown through a validation approach that the temperature and salinity measurements by our buoy systems are accurate, and capable of showing details of an upper ocean eddy during the Arctic winter. Additionally, post-deployment calibration only showed minor sensor drift. The dataset is expected to be of significant value for example for future process
studies, even beyond MOSAiC, and for the validation of numerical models aimed to better understand the role of the crucially important though barely studied Arctic Ocean in the Earth's climate system.

On a final note, buoy 2019O1 was redeployed in the new ice camp of MOSAiC Leg 5 close to the North Pole in August 2020 as 2020O10. As of February 2022, this buoy is still circling close to Jan Mayen island, although the CTDs ran out of power a few months earlier. Since this deployment was not part of the array, we decided to not include it in this paper. However, we will apply a similar processing to this dataset for consistency, as well as to future deployments of similar buoys that are are already planned for 2022, 2023 and 2024.

## 7  Code and data availability

Collections of the presented raw and processed datasets are available under https://doi.pangaea.de/10.1594/PANGAEA.937271 (Hoppmann et al., 2021i) and https://doi.pangaea.de/10.1594/PANGAEA.940320 (Hoppmann et al., 2022i), respectively (both currently under moratorium). An overview of all individual raw datasets (including unprocessed buoy- and CTD data) as well as all fully processed and quality-controlled datasets is given in Table 3). A token for temporary access to the reviewers can be generated at *here* for the processed data and *here* for the raw data. The Matlab code used to create the processed dataset is available on request.

*Author contributions.*  BR was responsible for the initial conceptualization of the study. The methodology was proposed by BR, and further enhanced by MH, YCF and IK. BR and MH were responsible for the project administration. MH and YCF deployed the instruments in the field. MH was responsible for data curation, prepared the original draft and visualized the data. IK conducted the data validation and eddy studies. All authors contributed to the review and editing of the manuscript.

*Competing interests.*  There are no competing interests associated with this study.

*Acknowledgements.*  Data presented in this paper were produced as part of the international Multidisciplinary drifting Observatory for the Study of the Arctic Climate (MOSAiC) with the tag MOSAiC20192020 (grant numbers AWI_PS122_00 and AFMOSAiC-1_00). We thank Andy Sybrandy and Pacific Gyre for building the buoys. The study was partly funded through the PACES II (Polar regions And Coasts in the changing Earth System) programme and the FRAM (FRontiers in Arctic marine Monitoring) strategic investment by the Helmholtz Association. The instruments were funded as part of the MIDO (Multidisciplinary Ice-based Drifting Observatory) infrastructure. The study contributes to the the Changing Arctic Ocean (CAO) program, jointly funded by the UKRI Natural Environment Research Council (NERC) and the BMBF, project Advective Pathways of nutrients and key Ecological substances in the ARctic (APEAR) grants NE/R012865/1, NE/R012865/2 and #03V01461; the project EPICA in the research theme MARE:N - Polarforschung/MOSAiC funded by the German Federal Ministry for Education and Research with funding number 03F0889A; and the European Commission for EU H2020 grant no. 101003472 (project Arctic PASSION). We thank the crews of the research vessels Akademik Fedorov, Akademik Tryoshnikov and RV *Polarstern*, as well as the helicopter company Naryan-Marsky for their great logistical support that made this study possible. We explicitly like to thank Kathrin Riemann-Campe and Marcel Nicolaus for their help with the intermediate data provision via meereisportal.de, and



Wilken-Jon von Appen for his input on the introduction. We express our gratitude to all participants of these expeditions, particularly to the MOSAiC School for their field assistance during the set-up, to Janin Schaffer and the Leg 3 team for their effort to collect the validation data, and to Matthew Shupe, Julia Regnery, Kirstin Schulz and the entire Leg 4 Team for the recovery of the instruments. Full acknowledgments are available in Nixdorf et al. (2021).





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

## Appendix A: Additional Figures & Tables

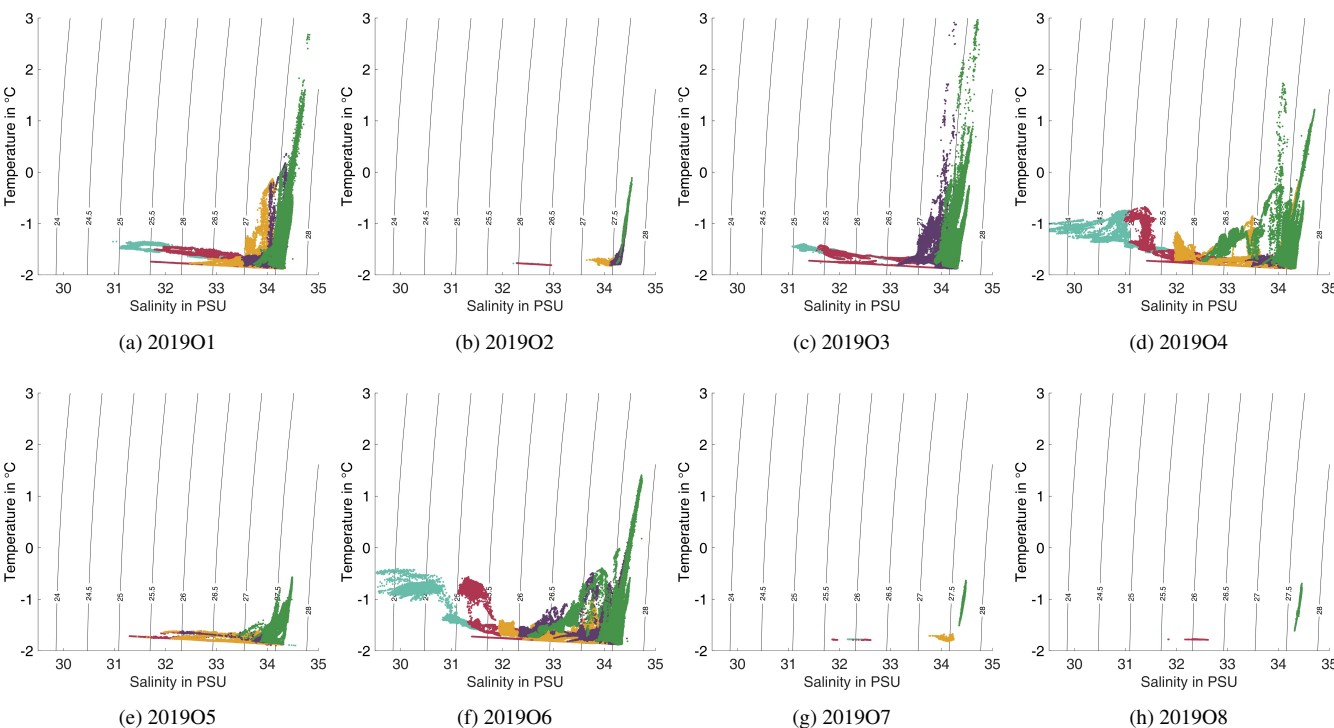

**Figure A1.** T-S diagrams based on the processed datasets collected by all eight buoys during their drift through the central Arctic as part of the MOSAiC Distributed Network. Note that these diagrams strongly depend on the operation lifetime of the corresponding buoy.





**Table A1.** Comparison of SST-CTD profiles with buoy measurements for validation.

| Station | Event (date) | Depth (m) | Temperature diff. | Salinity diff. |
|---------|--------------|-----------|-------------------|----------------|
| 2019O1 | PS122/3_38-101 | 10 | 0.0 | -0.012 |
| | 2020-05-02 | 20 | -0.002 | -0.012 |
| | | 50 | 0.003 | -0.006 |
| | | 75 | -0.001 | -0.012 |
| | | 100 | -0.004 | -0.017 |
| 2019O3 | PS122/3_37-119 | 10 | -0.002 | -0.002 |
| | 2020-04-26 | 20 | -0.001 | 0.004 |
| | | 75 | -0.001 | -0.016 |
| | | 100 | -0.002 | -0.005 |
| 2019O4 | PS122/3_38-102 | 10 | 0.0 | -0.002 |
| | 2020-05-02 | 20 | -0.001 | -0.004 |
| | | 50 | -0.001 | 0.005 |
| | | 101 | -0.001 | -0.003 |
| 2019O5 | PS122/3_37-120 | 10 | -0.002 | -0.002 |
| | 2020-04-26 | 20 | -0.001 | -0.001 |
| | | 50 | -0.004 | -0.002 |
| | | 100 | -0.004 | -0.005 |
| 2019O6 | PS122/3_38-103 | 9 | -0.003 | -0.006 |
| | 2020-05-02 | 19 | -0.003 | -0.001 |
| | | 50 | -0.004 | -0.004 |
| | | 75 | -0.004 | -0.002 |
| | | 100 | -0.002 | 0.001 |