# Peer review of "Mesoscale observations of temperature and salinity in the Arctic Transpolar Drift: a high-resolution dataset from the MOSAiC Distributed Network"

_Earth System Science Data, 2022_

## Author Response (AR1)

**Response to reviewer for Paper #ESSD-2022-66**

We again thank the reviewers for their constructive suggestions to our manuscript. The reviewer's concerns and advice guided us in re-evaluating the presentation of our results, and helped us to achieve a better understanding and (hopefully) to improve our manuscript.

In this revised version, we have added information and details to supplement the missing points raised by the reviewers. The introduction is substantially trimmed and the text relating to the submesoscale processes is moved to the later Section 5.3 for a more organized discussion. A newly added Figure 8 is now in the revised manuscript to strengthen our argument for potential submesoscale studies. We feel that our paper significantly improved in this process and hope that the reviewers and editor will agree. Thank you very much for your input, effort and guidance!

In the following sections we address the comments from each of the review accordingly. We have listed each comment followed by our response (in red text). We have also given the new line numbers in the revised text where we have made changes in accordance with the suggestions.

Review of Hoppmann et al., "Mesoscale observations of temperature and salinity in the Arctic Transpolar Drift: a high-resolution dataset from the MOSAiC Distributed Network" submitted for publication in Earth System Science Data.

**Comments from the reviewer**
**Reviewer #1**

Major comments:

This is a solid manuscript that I recommend for publication after mildly major revision. The writing and figures are generally very clear. There are some confusing explanations that could use some attention.

We thank the reviewer for the positive feedback. We tried our best to improve the explanations based on the detailed and very much appreciated comments. We hope that the new version is much clearer.

My major comment concerns a disconnect that I see between the Introduction, which has a significant discussion of the submesoscale, and the rest of the paper, which lacks analysis on these scales. See my comments near the end. The manuscript's title has the word "mesoscale" and the eddy analysis supports this. But why is there so much discussion of the submesoscale in the Introduction?

This issue was also rasied by another reviewer, and now we agree with this. Presumably this is because for quite a while we weren't sure how much science should go in the paper, and in the end there was a mismatch between the introduction and the preliminary analysis. To respond this comment, we have deleted a significant part of the introduction relating the submesoscales. We addressed the submesoscale as an added value of our dataset in Section 5.2 A newly added Figure 8 is provided in the revised text to illustrate the mesoscale aspect of the dataset and how it may be applied to study the submesoscale processes in the future.

Line-by-line comments

Line 61: Perhaps change "mesoscale scales" -> "mesoscales"

Agreed. We have fixed it. (Line 48)

Lines 93 and 101: "upstream in the Transpolar" ie insert "in"

Done. (Line 80; Line 88)

 Line 101: "2019/20" -> "2019", yes?

Done. (Line 88)

Line 114: cut the word "already."

Done. (Line 101)

Line 116-117: "the measurements ensued away from the inflow of warm Atlantic Water through the Fram Strait and as close as possible to the Siberian continental slope" I am not sure what you mean by this. Also, I suspect that "ensued" might not be the best choice of verb.

The expedition was meant to capture local processes at different locations within the Transpolar Drift system. The drift originates on the Siberian shelves and reaches to the Fram Strait and the Canadian Arctic Archipelago. The conditions contrast those parts of the Nansen Basin dominated by the relatively warm and salty inflow of Atlantic origin, where conditions near the surface are significantly less saline than in the Transpolar Drift. We have rephrased the sentence to better explain this idea. (Lines 102 – 108)

Line 119: "to achieve that aim" What aim?

The focus was to measure conditions and resolve processes in the upper mixed layer, seasonally deepening and subsequently restratifying, as well as the underlying stratification (upper halocline). The text had been clarified. (Lines 107 – 108)

Line 129: An egg has a pointy end; this buoy hull looks more oval, or perhaps "flattened oval"?

Agreed. We had changed "egg-shaped" to "oval". (Line 118)

Line 133: Your buoy naming convention is a bit odd, ie the use of the capital letter "O" instead of the number zero "0." But it is your choice. As noted below, this gets sloppy in the section about the eddy, when both "O" and "0" are used.

We hoped that the sloppiness has been corrected. The naming convention of these buoys is rooted in the larger context of the AWI buoy programme and the MOSAiC expedition, in which all the different buoy types are identified by the deployment year, a designated buoy-type dependent letter, and a running number. Actually, we should have included this description in the Methods part in the first place, which we have done now: "Following the naming convention of the general AWI buoy programme, which was also implemented for all MOSAiC buoys, we identify the individual buoys by an ID consisting of the deployment year (in this case "2019"), followed by a buoy type-specific letter (in this case "O") and a running number (here, "1" to "8"). This resulted in buoy IDs 2019O1 to 2019O8." (Lines 121 – 126)

Paragraph starting at Line 135: I got a bit confused here. I thought perhaps the CTDs recorded 2-minute data, and then every 10 minutes, all of these data were sent to the satellite. I think instead, the satellite gets only a subsample of the 2-minute data, i.e., 10-minute sampling, is this correct? I wonder if you can make this clearer.

This is partially correct. The buoy doesn't transmit a subset, it actually takes an additional measurement. We rewrote the paragraph substantially to make it clearer. (Lines 127 – 134)

Lines 153-158: This material confused me. The CTDs have pressure accuracy of 0.02 - 0.002 dbar, but the transmission limitation degraded this to 0.1 dbar. So this is a significant reduction in data quality, right? Your text seems to indicate that it's no big deal.

An accuracy of 0.1 \unit{dbar} implies a potential error of max. 0.09 \unit{dbar}, given the deprecated pressure values. That kind of change in the top 100 \unit{m} effects negligible changes in practical salinity and potential temperature (likewise, conservative temperature and absolute salinity). The error in salinity and potential temperature is always 10^-4 or less (using the corresponding standard units). The maximum error in depth is 0.09 \unit{m}, which, again, is negligible, given that standard CTD profiles by highly accurate systems are usually averaged at 1 \unit{m} or 1 \unit{dbar} intervals. The basis of these calculation is the Gibbs Seawater Toolbox, based on TEOS10.

While the deprecation of pressure to one decimal place is unfortunate and was not known to us when ordering the units, the problem will be alleviated by the manufacturer in any future production of those units.

Further, Table 2 indicates that the pressure accuracy is always 0.1 dbar, even for the CTD data. Why? I thought a direct download could give you 0.02-0.002 dbar. I guess I'm missing something.

Well, you are correct. The 0.1 is the most conservative accuracy, but yes, not representative for most of the dataset. We changed this to 0.1 / 0.02*, with an additional comment in the table caption. We would prefer not to give any higher accuracy, because the availability of pressure sensors was limited at the time when the CTDs were built, so for example 1000m rated pressure sensors ended up in some instruments.

Line 191: "an upper threshold of 0.8 m s-1 was applied" Why?

Actually, no single drift speed data point was removed based on that threshold. So this statement was removed to not confuse the readers. (Lines 189 – 191)

Line 220: what is "sim"? I suspect it is some kind of word processing code.

We apologize for this formatting issue. It should have been \sim to denote the LaTeX code for ~. The formatting is now correct, showing ~10 months in the revised text. (Line 220)

Line 222: How do you define "suspicious?" and Line 227: How do you define "questionable?"

Beyond the defined range checks and moving average filters, a manual spike flagging was performed that was not based on a strict and stiff definition, but rather on a subjective (expert) judgement to the best of our knowledge. Examples are obviously isolated spikes (that still didn't fail the other tests, and therefore can't be considered "obvious" outliers), or suspicious "episodes" that for example weren't accompanied by concurrent changes from any other parameters.

In that context, there are other flagging schemes that even distinguish between suspicious and questionable (which we don't), or that categorize into "probably good" and "probably bad" etc. We don't feel comfortable in using that level of differentiation, and therefore decided to flag everything that "looks odd" or inconsistent with the -questionable- flag.

Line 233: "When CTD data was available…" This caused me some confusion, because the buoy has CTDs. Perhaps you can add a sentence like this to make it clearer: "Buoy data" means all data uploaded to the satellite from the buoys, including from the CTDs, at 10 min sampling, while "CTD data" refers to the directly downloaded CTD data from recovered buoys at 2 min sampling. Is that right?

This is correct. We had revised the text, also based on the feedback by reviewer 3 who had the same issue. (Lines 234 – 235)

Table 2: Is the surface temperature thermistor really accurate to 0.01 degC? I think these are usually not so good, e.g., 0.1 or 0.05 at best.

This is correct, this should have read "0.05", and has been changed accordingly.

Line 395: The MLD is not evident to me from this figure. How was it computed?

see comment below on MLD in Figure 4.

Line 397: Perhaps change "a corresponding decrease in temperature" -> "a corresponding decrease in temperature along the freezing line"

Good suggestion, the text has been changed accordingly. (Lines 411 – 412).

Lines 399-400: Perhaps change this text to: "...partly explained by a combination of upward mixing of deep salty water from below and salt rejection during ice formation from above, both forced by two February storms (with wind speeds up to 16 ms-1 and air temperatures of XX degC)."

Good suggestion, the text was changed accordingly. (Lines 412 – 415)

Lines 400-401: why is "vertical mixing properties" in quotation marks?

The quotation marks are not necessary. This has been corrected. (Line 415)

Line 403: I can't see MLD in Figure 4.

This is correct, a continuous MLD evolution naturally can't be derived from our fixed-depths dataset. However, when CTD records from different depths have similar or nearly identical readings, one can infer that the MLD has deepened at that time (i.e. the upper 100m have the same physical properties). (Lines 417 – 418)

Line 404: What is "air-line distance?"

This was changed to "straight line distance", which is hopefully more clear. (Line 418)

Line 413: "The position of the eddy remains approximately the same" The same as what?

We thought this sentence did not yield useful information and could be speculative. This sentence has been removed in the revised text.

Lines 417 & 419: check your velocity units. m-1 is wrong, yes?

We apologized for this mistake. This has been corrected. (Lines 434 – 437)

Line 417: "estimated diameter of the eddy" estimated how?

The estimate was based on the buoy drift distance. We had added more details in the revised text. (Lines 434 – 435)

Line 421: Here you are using zero "0" instead of capital "O." There are several other examples in this eddy section.

Thank you. We had corrected this inconsistency in this section.

Line 426: "by a factor of" i.e., add "a"

We had decided to express the buoyancy frequency in a unit of cph (cycle per hour) so it may be physically more intuitive. Since the unit is modified so we no longer need to emphasize "by a factor of 2". The text is revised accordingly. (Lines 444 – 445)

Line 428: "radii" -> "radius"

Fixed. (Line 446)

Lines 431-432: Great point!

This is what we want to contribute to the community. Thank you.

Line 439: "measured the the upper ocean quasi-synoptically over submesoscales" Two comments: (1) cut one "the" (2) Your introduction has a lot of material about submesoscales, but the paper itself has no corresponding analysis. I suggest you either cut / trim down the introduction material on this subject, or do more work to follow up on it. For example, you could create a histogram for one or all the buoys with a horizontal axis of del-x = the distance travelled by a buoy over 2 or 10 minutes. If you have a lot of sampling between 100 m – 10 km (or maybe shorter? Your introduction implies that Arctic submesoscales are shorter) then yes, you can justify this statement on Line 439 and the extensive discussion of this subject in the Introduction.

Thank you for this comment, it makes a lot of sense. We have strengthened the content in Section 5.3. As mentioned further up, we have removed the mismatch between the introduction and the analysis by moving the submesoscale material to Section 5.3, while also alongside this including a newly added Figure 8 that is in principle similar to the histograms the reviewer suggested. We hope that the match is much better now. (Lines 450 – 480)

But I would also ask this question: Did you really need to sample at 2 or even 10 minutes? What if you sampled hourly, or even daily? What would this histogram look like? I suppose I am suggesting that you do some analysis to find the right time interval for buoy data recording to adequately sample the submesoscale, as you claim without proof that you are doing now.

The choice of 2 and 10 minute sampling was based on a power budget calculation of the CTDs and buoys, and optimized towards the ideal duration of the MOSAiC experiment, which was one year.

Furthermore, resolving various processes, such as internal wave variability, requires sampling intervals of a few minutes. We have provided a newly added Figure 8 discussed on what time and space scales are resolved in our dataset in Section 5.3. (Line 450 – 464)

Line 473: code is available "upon request." Is this sufficient for this journal? Should it instead be available on github or equivalent?

Thank you for your comment. Since the processing code doesn't include any particularly innovative techniques and is only tuned towards this particular dataset, we decided that there is not enough added value to putting the project on Github or similar repositories. If this is a requirement for the journal though, we might reconsider this.

**Comments from the reviewer**
**Reviewer #2**

Major comments:

This is a solid contribution to the field, it does have some issue with grammar and word choice throughout the text, however, it did not hinder my ability to read or understand the content, maybe the authors could run it through a grammar checking program (I use Grammarly).

We thank the reviewer for the comment and advice. We now used Grammarly to correct the grammar. We also tried to improve the wording in some instances, although it is difficult to judge what words exactly the reviewer is referring to. We hope that the overall reading experience is better now.

From the title of the manuscript I expected the focus to be primarily on analysis of the data, however the focus of the paper is really on describing the buoy system and the handling of the data. I think that this does fit within the scope of the journal, but the authors may want to change the title to better fit with the focus of the paper. I have no issues with the substance of the paper, just a few comments to improve clarity.

We appreciate the comment and concern about the title "Mesoscale observations of temperature and salinity in the Arctic Transpolar Drift: a high-resolution dataset from the MOSAiC Distributed Network". It is usually quite difficult to represent an entire manuscript by only a few words. However, we are not entirely sure why the title would suggest a focus on a scientific analysis, as words like "observations" and "dataset" should indicate that the focus is on the description of the data. We could probably have written a more technical paper and submit to a more tech-focused journal, but we wanted to put the focus on the dataset itself due to its uniqueness, but with the technical part still properly represented. In short, we do believe that the title is appropriate in its current form, but we are absolutely open and willing to change it if the editor thinks this is necessary.

The description of the data handling and QC is well thought out and helpful for others who are starting out with automated buoy systems.

We appreciate the positive feedback.

Line-by-line comments

Line 135. I was a bit confused by the statement "polled for an additional measurement by the buoy itself" I think that you mean that the CTDs recorded data every 2 mins and then they were also collected a measurement every 10 mins that was sent via iridium. This paragraph is not very clear.

The other reviewers also criticised that this part was not very clear. The paragraph has been completely rewritten as follows:

"In order to ensure an operational time of ~1 year, the individual CTDs were set to record data internally at 2 minute intervals, independent of the buoys' own sample and transmit intervals. The surface buoy itself recorded GPS position, surface temperature, and carried a submergence sensor. Furthermore, the buoy controller polled all CTDs for an additional measurement independent of the CTDs' internal sampling according to a pre-configured buoy sampling interval, which could in principle be adjusted by sending a reconfiguration command via iridium if necessary. However, throughout our experiment, all buoys were set to take a measurement and immediately transmit the corresponding data at a fixed 10 minute interval, chosen to ensure an operational time of at least one year. Thus, the internal sampling interval of the CTDs was 2 minutes, while an additional CTD measurement was obtained and transmitted by the buoy every 10 minutes together with the corresponding auxiliary (meta)data." (Line 127-134)

Line 404: 'air-line distance' do you mean the the straight line distance rather than the distance along the drift track.

This is correct. We changed it to "straight line distance". (Line 418)

Line 404: Is there a better way to visualize the MLD in Figure 7?

Actually, a continuous MLD evolution can't be derived from our five fixed-depths dataset. However, when CTD records from two or more different depths have similar or nearly identical readings, one can at least infer that the MLD had deepened at that time. At the current stage, we don't have a better way to visualize the MLD in Figure 7.

**Comments from the reviewer**
**Reviewer #3**

Major comments:

The authors present a dataset of upper ocean temperature and salinity from drifting buoys that were deployed as part of the MOSAiC Distributed Network. The dataset represents valuable observations from the central Arctic Ocean over a 10-month period, including the scarcely observed Arctic winter. Observational methods, data processing steps and the resulting dataset(s) are described in detail, and some preliminary analyses of the data to investigate the signature of an eddy in the distributed network are shown.
The manuscript is in general well written with mostly clear and useful figures. Some revisions could help to improve readability and avoid confusion in an otherwise highly relevant publication.

We thank the reviewer for the positive feedback. We incorporated the comments into our revised manuscript, hoping to remove the confusion and improve the overall quality.

There is a bit of repetition between the Methods and the Results section, esp. with 4.1. This is maybe not surprising given the nature of a data paper, but could still be avoided, and the manuscript could be more streamlined.

We agree with this assessment, but this is somewhat intentional. This dataset as a whole is a little complicated, and there is great potential that a reader might get confused easily if the manuscript is only partially read. We intentionally included some repetitions where appropriate to make an individual section (such as 4.1) as standalone as possible, thus the readers do not need to read the whole sub-sections in detail. There is absolutely a lot of room for us to trim the text and make the manuscript more concisely. For the revised text we would prefer to maintain this writing style, although it may make the overall manuscript a bit longer. We hope that this consideration makes sense to you.

The introduction is quite lengthy about mesoscale and submesoscale features, but then there is nothing more these throughout the methods and the results section. First in the discussion, you pick up the topic again by describing the passage of an eddy. To show that the dataset is useful to investigate (sub)mesoscale features, which you explicitly present as an aim of the design of the DN and this manuscript, a bit more analysis in this direction would be useful. For example, more detail on the drift trajectories of the buoys relative to each other beyond just the map in Figure 3 (and the short sentence on Lines 267-269) could easily provided, such as a timeseries of the relative distance of the buoys to each other or of the maximum diameter of the area covered by the buoys. That would link nicely to the discussion around the size of the eddy and could indeed demonstrate that this buoy network can capture (sub)mesoscale features.

As also explained in our response to reviewer 1, we had tried to improve the previous mismatch between the introduction and the analysis by moving some of the submesoscale material to the discussion section (i.e., Section 5.3 is substantially strengthened). This is alongside with a newly

added Figure 8, which shows the TS diagrams with different sampling intervals applied (which is analogous to your suggested idea of the distance analysis, please see Lines 450 – 480).

Upon the submission of this manuscript, there are currently 3 more manuscripts in preparation that use the present dataset to look at (sub)mesoscale features. Thus, it is beyond the scope of this paper to dig into too many details. We hope that our trimmed introduction and strengthened Section 5.3 (and Figure 8) are sufficient to respond your comments.

Be more precise with the variable names – I have to assume that you talk about in situ temperature and practical salinity but cannot know for sure, so please make this clear at the start of the manuscript.

This is correct. There are indications about this throughout the manuscript, but now we state this explicitly at the beginning of the data processing section:

*"In this paper, we exclusively refer to in situ temperature (T in ∘C) and practical salinity (S in PSU), if not stated otherwise."* (Line 176)

Many of the figures are not colour-blind (or greyscale friendly) – consider modifying them.

Thanks for the suggestion, we are looking for colour-blind friendly palette and planning to improve this issue in the final version.

Line-by-line comments

Line 21: "much greater than" – can you be more precise? E.g. give order of magnitude or similar?

Changed text to indicate that timescales are up to the order of months. (Line 21)

Line 21&22: "the vertical velocity" and "the horizontal velocity" – sounds odd (like there is only one fixed vertical velocity and one horizontal velocity); suggest to change to "vertical velocities are … weaker than horizontal velocities".

Done. Line 22

Line 41: "submesoscale processes are responsible for … restratification" – suggest to change to "contribute to". Du Plessis et al. (2019) still state that the main driver for springtime restratification is surface heating.

As also suggested by other reviewers, we had moved most parts of the submesoscale materials from the introduction and discussed them in Section 5.3. Original text had been modified and is now in Section 5.3. There are many processes driving mixing and restratifying the water column, dependent on season, region etc. Du Plessis et al. (2019) name submesoscale processes as strongly influencing seasonal restratification processes, e.g. their abstract states: "*An increase of*

*stratification from winter to summer occurs due to a seasonal warming of the mixed layer. However, we observe transient decreases in stratification lasting from days to weeks, which can arrest the seasonal restratification by up to two months after surface heat flux becomes positive. This leads to interannual differences in the timing of seasonal restratification by up to 36 days."* The point of the citation in our manuscript is to put forward the importance of submesoscale processes for the upper ocean, not to give an accurate account of seasonality in the southern oceans. In the Arctic, horizontal density gradients can lead to submesoscale adjustment in the mixed layer, effecting restratification in the formerly well-mixed upper layer (e.g. Timmermans et al., 2012). (see our strengthened Section 5.3)

Line 66: Is "synopticity" really a word? (Or rather, a word in this context?) Suggest to rephrase.

This has been changed and now refers to the design of hydrographic surveys. (Line 53)

Line 84: Which feedback? Sounds like there is only one…

This has been changed to reflect that there are manifold feedbacks, without going into further details though. (Lines 71 – 73)

Line 93: Do you mean upstream IN or OF the Transpolar Drift?

We added an "in". (Line 80)

Line 101: Same issue: upstream in or of the TPD?

We added an "in" as well. (Line 88)

Line 109-110: "the ice relative drift across the liquid ocean" – weird formulation. Rephrase?

This has been rephrased to just "ice drift". (Line 96)

Line 117: "ensued" – is this the right word? Or should it rather be something like "took place" or "were done"? I assume there was a degree of planning involved in the location of the DN.

The introduction has been changed extensively, and the wording mentioned here was modified in the process. (Lines 102 – 108)

Line 135-138: This part is confusing. So the CTDs are measuring every 2 minutes regardless of what the buoy is doing. Then there's an extra measurement when the buoy is polling – so "its measurement interval" means the buoy's measurement interval? And the data the buoy sends back via iridium – that's only those extra measurements or all of them?

This paragraph has been strongly modified, as reviewer 1 had the same problems understanding it. We hope it is much clearer now. You are generally right, the buoy wakes up every 10 minutes and lets the CTDs take a separate measurement, regardless of the CTD internal recording interval. Only this one is transmitted. (Lines 127 – 134)

Following on from that, on line 142 you write "All data" – is that now the data from the measurements polled by the buoy or indeed "all" data??

"All data" in that case refers to the CTD readings as a result of the "extra" measurement, along with GPS, surface temp and submergence. The buoy does not have any means to access the internal storage of the CTDs.

Please rework this part to clarify.

The part was reworked as "*The data resulting from this measurement cycle was then compressed into a file and sent via the Iridium short burst data (sbd) protocol to a shore-based server, which decoded the data and provided the data for download.*". (Lines 138–139)

Line 145 and throughout: when you write "temperature", is that in situ temperature? Please clarify in the text.

we added a general sentence at the beginning of the data processing section: "In this paper, we are exclusively referring to in situ temperature (T in $^\circ$C) and practical salinity (S in PSU), if not stated otherwise." We hope this makes it clearer. (Lines 176 – 177)

Line 147: "the other variables" – I assume that's conductivity, (in situ?) temperature and pressure?

Yes, this has been made clearer now as "Salinity was calculated on the server side based on the transmitted temperature, conductivity and pressure measurements according to the Practical Salinity Scale 1978 (PSS-78). For the final dataset however, practical salinity was recalculated from these variables using the Matlab GSW toolbox version 3.05.5 (McDougall and Barker, 2011).". (Lines 141 – 144)

Line 147 (and throughout the manuscript) provide version numbers of software and toolboxes used.

Version numbers have been added where relevant. (Line 144)

Line 169: Protection from what? Towels don't strike me as the best protector from freezing in -15deg C.

Yes, you might be right that towels are not the best protection against cold air temperatures. There were significant constraints on logistics and the helicopter was full, with a tight flying window, which led to us not having any good means to keep the instruments heated pe-deployment. It might be embarrassing, but in the end we had to use towels to at least provide some sort of protection while handling the CTDs on the ice and in the snow. We deployed the tether *manually* with only 2 persons without the help of a tripod, so the instruments had to be placed on the ice/in the snow for up to an hour while we attached them to the cable. This also

means that there were no 5 persons to handle every instrument while lowering the tether in the hole, so we actually had to drag them over the ice. To make a long story short, we did only have towels, which is better than nothing, and next time it should probably be done differently.

Line 171: What do you mean by "hydrohole"?

This is the term for an access hole through the ice into the ocean, and is regularly used in the literature. We changed this now to "hole in the ice" though. (Line 169)

Line 179: The CTD data from recovered instruments then also provide higher accuracy pressure data?

This is correct. As described further down in the data processing section, we even used the more accurate recovered CTD data to increase the accuracy of the corresponding transmitted buoy data (A step called "Merge dataset" in Figure 2).

Line 186: Replace "using" with "indicated by" or something similar (you didn't "use" the flag to modify the data).

Changed as suggested. Line 185

Line 190: How did you fill the gaps in the GPS record after removing outliers?

The gaps were filled by linear interpolation. This has been added. (It was just a handful of outliers.) (Line 189)

Line 191: What was the consequence of applying the threshold? A different flag? Or removing data points?

Actually, no single drift speed data point was removed based on that threshold. So this statement was removed to not confuse the readers. (Lines 189 – 191)

Line 192: How did you determine "plausibility"?

We just did a general range check and looked at the seasonal evolution. We did not compare or validate against other meteorological data, since this parameter is not in the focus.

Line 220: What do you mean by "sim"?

A formatting issue in the LaTeX. This should have been \sim, which is the LaTeX code for ~. It is fixed now. (Line 220)

Line 223: How big was the drop in conductivity?

The drop in C was from ~34.1 to ~33,5 PSU, and not accompanied by a temperature change. This information has been added. (Line 223)

Line 229: Capital O in the buoy name.

Fixed. (Line 230)

Line 240: Add "practical" to salinity.

Done. (Line 241)

Line 243: Is the closing bracket in the right place? Bit confusing at the moment.

The comma was in the wrong place, and it should be clearer now. (Line 244)

Line 253-254: It would be useful to include this information in one of the tables, e.g. Table 1. If you explicitly state here all buoy numbers of the ones you can provide a merged product from, then also provide the numbers of the ones with buoy data only.

Good point. This information has been added to Table 3, where it is most relevant. Also, we included a detailed overview to the part this comment refers to.

Line 255: I guess the measured temperature is in situ temperature? And the derived salinity practical salinity (and not absolute salinity)?

This is correct, and has been added here. We also added a general sentence as suggested in another comment. (Line 253-254)

Figure 2: Nice schematic! You could mark which of the steps were not included in the processing for buoy-only datasets.

Actually this is the purpose of the steps that have a similar grey background as the SBE37 field at the top. Those have not been applied to the buoy-only datasets.

Table 2: In situ temperature?

Yes, this has been added.

Figure 5d: axis labels in the salinity panel are missing. If they are not included because of the icing issues, why show the data at all?

Thanks for catching that, the labels should be there. The graph shows the reasonable data. This will be corrected in the next version.

Presentation of the timeseries starting Line 274: Refer to the respective figures/panels.

References have been added to the time series description accordingly. (Lines 273 – 330)

Line 276: delete "a"

Thanks. The text had been revised. (Lines 275 – 276)

Line 276-277: Isn't 2.5 weeks a pretty long time for ice in the conductivity cell to disappear? Was the water at freezing point at 50 m depth throughout this period?

We tend to agree, but we don't have any other explanation for that effect. The salinity was completely off track and did not suggest any ambient adjustment from the water column. We included a sentence that this is rather unexpected, as you suggested. (Lines 275 – 276)

Line 286: So all but the 50 m sensor?

The 50m sensor didn't give any data, so we can't refer to any data explicitly. But since all other 4 sensors showed the same consistent behaviour, we conclude that the events described must have been affected the tether as a whole. We considered to add a sentence along those lines to the manuscript, but instead changed "all CTDs" to "all remaining CTDs", which we consider sufficient. (Lines 285 – 286)

Line 318-319: Is a drop in pressure not equivalent to instruments rising? "… causing the surface buoy to drop" => shouldn't that then lead to an increase in pressure? I can't follow your argument here.

Thank you for catching this point. Of course we meant that the pressure increased. This has been corrected. (Lines 321 – 323)

Line 323-324: How long did this issue last? The entire record?

Yes, this issue remained for the entire operational period. Note that this buy was anyways operational for only two weeks. This information has been added. (Lines 327 – 328)

Line 329: Replace "in" by "the".

Done. (Line 334)

Line 338: Where below? Provide section/section number.

Reference to Section with description of cross-validation has been added. (Line 347)

Line 374: What is the stated accuracy?

The stated accuracies have been added. (Lines 383 – 385)

Figure 7: I'm confused by the shading in panels c and d – please provide a clearer description in the caption.

We agree with this comment. The shading is now removed in Figure 7c and 7d, which are revised by using dashed lines and horizontal bars to highlight the time ranges we want to show.

Line 472: Publish the code together with the dataset or place it on e.g. github or similar sites.

Reviewer 1 had the same comment. We did consider it, but since the processing code doesn't include any particularly innovative techniques and is only tuned towards this particular dataset, we decided that there is not enough added value to putting the project on Github or similar repositories. If this is a requirement for the journal though, we will of course reconsider this.

Figure A1 is not referred to in the text.

This figure has now been referred to in the main text. (Line 270)

---

## Author Response (AR2)

**Response to reviewer/editor for Paper #ESSD-2022-66**

We thank the reviewers and the editor for their constructive suggestions for our manuscript. As outlined in our response document from 22 July, the reviewer's concerns and advice from the first review round guided us in re-evaluating the presentation of our results and helped us immensely to improve our manuscript. We are also very grateful to the Editorial Support Team, who has helped us a lot during the review process.

In his latest review, the topical editor requested a minor revision of the Abstract:

"A small addition. The referees have noted that 'mesoscale' is the main issue. This concept should be included also in the abstract (actually it isn't). I suggest to add a short sentence (line 6) before 'The multi-sensor system ...'. something like: Although the data sets can contribute to mesoscale and sub-mesoscale processes' studies, this paper is mainly focused on the mesoscale.

Thank you for this suggestion. We added a similar sentence to the end of the Abstract, where we believe it fits best. "While this dataset also has the potential to contribute to submesoscale process studies, this paper mainly highlights selected preliminary findings on mesoscale processes."

"About the code availability, I agree with you: there isn't any innovative technique. It can be available under request."

Yes, we agree and will keep it available on request.

In addition to these points, we have also implemented additional grammatical improvements based on Grammarly (as was suggested by a reviewer earlier), which hopefully further improves readability.

Finally, we modified the problematic Figures 4, 5, and A1 to be more color-blind friendly.

The last change we need to implement after acceptance of the manuscript is to adjust the links to the PANGAEA datasets, where the DOIs still need to be registered once the review process is successfully concluded. We hope that's ok.

Cheers, Mario, Ben, Ivan, and Fang